# Dysregulation of MMP2-dependent TGF-ß2 activation impairs fibrous cap formation in type 2 diabetes-associated atherosclerosis

Pratibha Singh[1,8], Jiangming Sun [1,8], Michele Cavalera[1], Dania Al-Sharify[1], Frank Matthes [1], Mohammad Barghouth[1], Christoffer Tengryd[1], Pontus Dunér[2], Ana Persson[1], Lena Sundius[1], Mihaela Nitulescu[1], Eva Bengtsson[2,3,4], Sara Rattik[2], Daniel Engelbertsen[2], Marju Orho-Melander [2], Jan Nilsson [2], Claudia Monaco [5], Isabel Goncalves[1,6,9] & Andreas Edsfeldt [1,6,7,9] ✉

Type 2 diabetes is associated with cardiovascular disease, possibly due to impaired vascular fibrous repair. Yet, the mechanisms are elusive. Here, we investigate alterations in the fibrous repair processes in type 2 diabetes atherosclerotic plaque extracellular matrix by combining multi-omics from the human Carotid Plaque Imaging Project cohort and functional studies. Plaques from type 2 diabetes patients have less collagen. Interestingly, lower levels of transforming growth factor-ß distinguish type 2 diabetes plaques and, in these patients, lower levels of fibrous repair markers are associated with cardio-vascular events. Transforming growth factor-ß2 originates mostly from contractile vascular smooth muscle cells that interact with synthetic vascular smooth muscle cells in the cap, leading to collagen formation and vascular smooth muscle cell differentiation. This is regulated by free transforming growth factor-ß2 which is affected by hyperglycemia. Our findings underscore the importance of transforming growth factor-ß2-driven fibrous repair in type 2 diabetes as an area for future therapeutic strategies.

Individuals with type 2 diabetes (T2D) are at high risk of suffering from both cardio- and cerebrovascular complications (such as myocardial infarctions and ischemic strokes) and premature death due to an early aggressive atherosclerotic disease. Considering the increasing prevalence of T2D, the prevention of cardiovascular (CV) complications among individuals with T2D has become one of today's greatest challenges in medicine.

The aggravated atherosclerotic plaque formation in T2D, leading to a greater plaque burden and more frequent plaque ruptures, is well described, and despite our clinical advances in treating cardiovascular

disease, patients with T2D and CV risk factors still have a clearly increased risk of suffering from cerebro- or cardiovascular events[1–5]. This suggests that we are missing important biological processes in T2D atherosclerosis, which could be targeted to prevent future complications.

Plaque matrix metalloproteinases, and especially matrix metalloproteinase (MMP)9, induced extracellular matrix (ECM) degradation, triggered by pro-inflammatory transcription factors such as NF-κB or c-Myc, have been extensively studied and suggested as key processes in plaque ruptures[6–8]. Yet, recent human plaque studies have not been

[1]Cardiovascular Research–Translational Studies, Lund University, Malmö, Sweden. [2]Department of Clinical Sciences Malmö, Lund University, Malmö, Sweden. [3]Department of Biomedical Science, Malmö University, Malmö, Sweden. [4]Biofilms–Research Center for Biointerfaces, Malmö University, Malmö, Sweden. [5]Kennedy Institute of Rheumatology, Nuffield Department of Orthopaedics, Rheumatology and Musculoskeletal Sciences, University of Oxford, Oxford, UK. [6]Department of Cardiology, University Hospital of Skåne, Lund/Malmö, Sweden. [7]Wallenberg Centre for Molecular Medicine, Lund University, Lund, Sweden. [8]These authors contributed equally: Pratibha Singh, Jiangming Sun. [9]These authors jointly supervised this work: Isabel Goncalves, Andreas Edsfeldt. ✉e-mail: Andreas.edsfeldt@med.lu.se

able to confirm an enhanced inflammatory activity, as measured by the presence of CD68[+] macrophages or plaque levels of pro-inflammatory cytokines, in human T2D plaques[9,10]. Instead, we have provided evidence for changes in ECM composition as an important biological factor contributing to the formation of high risk plaques in T2D[9]. This hypothesis has been supported by imaging studies which repeatedly have identified thin fibrous caps as an important characteristic of T2D atherosclerotic plaques[11–14]. The thin cap fibro-atheromas, with reduced ECM proteins (mainly collagen), are known to be rupture-prone and associated with clinical events.

ECM composition is tightly regulated by its synthesis and degradation and depends on the delicate interplay between smooth muscle cells and immune cells. Beyond the degradation of ECM, the regulation of its synthesis and repair has been much less characterized in human atherosclerosis so far.

In this work, we mapped the cellular, molecular and spatial features of the ECM biology of the human atherosclerotic plaques that distinguish plaques from patients with and without T2D and revealed a key role of TGF-ß2-driven vascular smooth muscle cell (VSMC) biology in patients with T2D.

## Results

### Lower levels of collagens are associated with future CV events in T2D

To explore if plaques from patients with T2D have smaller protective fibrous caps, cap area was assessed histologically. As indicated by imaging studies, T2D plaques had significantly smaller fibrous cap areas (3,665,613 (IQR 2,167,000–5,379,269) $\mu m^2$ vs 6,252,125 (IQR 4,667,931–10,673,356) $\mu m^2$; $p = 0.0037$; Fig. 1a) and smaller cap areas in % of plaque area compared to non-diabetes (ND) plaques (14.5 (IQR 11.4–18.0)% vs 25.5 (IQR 18.6–34.0)% of plaque area; $p = 0.0008$; Fig. 1a). As collagens are the predominant proteins in the cap, plaque collagen content (from homogenates) and collagen area in the most stenotic plaque region were measured. Importantly, both collagen plaque levels (as previously shown in smaller cohort)[9] and collagen plaque area were reduced in T2D compared to ND plaques (40.1 (IQR 31.2–50.1) vs 54.2 (IQR 34.4–74.9) mg/gram wet weight plaque, $p = 2.0 \times 10^{-4}$ and 1.9 (IQR 0.9–3.7) vs 2.4 (IQR 1.1–7.3)% of plaque area, $p = 0.039$; respectively; Fig. 1b, c).

Next, we explored if an imbalance between markers associated fibrous matrix repair and markers associated with a vulnerable plaque phenotype predicted future CV events among T2D patients using a previously published histological vulnerability index (VI)[15,16]. Among T2D patients, plaques with a high VI (3rd tertile compared to 1st–2nd) were associated with a higher risk for future CV events (Fig. 1d). Importantly, a multiple linear regression model identified smaller collagen and alpha-actin[+] plaque areas as the two factors with the strongest associations to the VI in T2D (Fig. 1e, $p = 0.04$ and 0.001 respectively).

### Lower levels of TGF-ß discriminate plaques from patients with T2D

As the collagen formation associated with fibrous repair is dependent on growth factors, we explored if differences in plaque levels of growth factor could separate T2D from ND plaques. Using a panel of 16 growth factors, an orthogonal partial least squares discriminant analysis (OPLS-DA) analysis was performed, which revealed that T2D plaques selectively clustered from ND plaques (Fig. 2a) with goodness of fit and prediction measured by R2Y (cumulative) = 0.19 and Q2 (cumulative) = 0.14 and CV-ANOVA ($p = 1.1e$-6; Supplementary Fig. 1). Overfitting was not observed as shown in further permutation tests (Supplementary Fig. 1). Further, VIP statistic suggested TGF-β3 and TGF-β2 as key factors separating T2D plaques from ND plaques (Fig. 2b).

To corroborate the findings from our multivariate analysis, we tested if TGF-β levels differed significantly between the two groups.

Interestingly, plaques levels of all three TGF-β isoforms (-β1 126 (IQR 74–266) vs 239 (IQR 116–409) pg/g wet weight plaque, $p = 2.0 \times 10^{-4}$, -β2 1004 (IQR 630–1454) vs 1485 (IQR 898–2843) pg/g wet weight plaque, $p = 1 \times 10^{-5}$, -β3 40 (IQR 23–64) vs 72 (IQR 43–126) pg/g wet weight plaque, $p = 1 \times 10^{-8}$) were lower in T2D compared to ND plaques (Fig. 2c). The relative amounts are in line with what has been previously described[17]. In line with this, in vitro studies of TGF-β release from isolated live human plaque cells showed reduced levels of all three isoforms in T2D cell supernatants compared to ND cell supernatants (β1 47,960 (SD 6920) vs 77,753 (SD 10,823) pg/g total protein, $p = 0.004$; β2 173,793 (SD 28,771) vs 878,234 (SD 105,780) pg/g total protein, $p = 1.4 \times 10^{-5}$; β3- 219 (SD 27) vs 918 (SD 69) pg/g total protein, $p = 0.7 \times 10^{-7}$ Fig. 2d).

Lastly, we explored if any of the assessed growth factors were associated with the calculated VI and found that TGF-β2 was the only isoform negatively correlated to the VI ($r = -0.36$, $p = 7 \times 10^{-7}$; two-stage step-up method of Benjamin, Krieger and Yekutieli adjusted $p = 4.0 \times 10^{-5}$).

### Contractile smooth muscle cells are the main sources of TGFB2

Due to the heterogeneous cellularity of the human plaque tissue, we next explored the cellular architecture of the human plaque to characterize the fibrous repair process and cellular source of TGF-ß, which have not been explored previously. To this end, single-cell deep RNA sequencing (ScRNAseq) data of live human carotid plaque CD45[+] ($n = 366$) and CD45[−] cells ($n = 489$) was performed (Fig. 3; gating strategy is provided in Supplementary Fig. 2). CD45[−] cells were the main source of TGFB2 and -B3 whereas TGFB1 was expressed evenly by a vast majority of plaque cells (Fig. 3a).

Furthermore, clustering analysis revealed five major CD45[−] plaque cell subgroups as shown in the t-SNE plot (Fig. 3b). Clusters 1–3 and 5 were represented by smooth muscle cells (ACTA2[+]) and cluster 4 by endothelial cells (PECAM1[+]; Fig. 3c and Supplementary Tables 1–5). Based on key cell phenotype markers and transcription factors involved in VSMC differentiation, cluster 1 was identified as contractile VSMCs and cluster 3 as synthetic/fibroblast-like VSMCs (Supplementary Fig. 3a, b). Top overexpressed genes (CD36, RGS5, KCNJ8, PDGFRB, THY1 and LPL, Supplementary Tables 1–5) also indicated that cluster 2 cells could be adipocyte-like VSMC enriched with the scavenger receptor CD36 as well as the LPL gene. Based on top marker genes, cluster 5 cells were identified as macrophage-like smooth muscle cells enriched in genes commonly associated to myeloid cells, including CCL5, FTL, LYZ and RGS1.

Of all CD45[−] cells from the ScRNAseq cell analysis, 24.5% were identified as contractile VSMC, 23.1% as adipocyte-like VSMC, 22.1% as synthetic/fibroblast-like VSMC and 11.5% as macrophage-like VSMC (Fig. 3a, b). Of all CD45[+] cells, 52.7% were T-cells, 12.6% were natural killer cells, 9.0% were dendritic cells and 25.8% were macrophages (macrophage 1, 18.6% and macrophage 2, 7.1%, respectively; Fig. 3a and Supplementary Fig. 4a, b).

To further validate that the four VSMC clusters were not fibroblasts, macrophages or chondrocytes, we investigated the overlap of genes commonly used as prototypical cluster markers for these cell types, but no major overlaps were found (Supplementary Fig. 5).

TGFB2 was expressed in a greater proportion of cluster 1 cells (proportion = 0.183; contractile VSMC) compared to cluster 2, 3 and 5 cells (proportion = 0.076, $p = 1.0 \times 10^{-4}$; Fig. 3d). A higher proportion of TGFB1[+] cells was also found in cluster 1 cells (proportion = 0.442) compared to clusters 2, 3 and 5 cells (proportion = 0.283, $p = 2.0 \times 10^{-4}$). TGFB3[+] cells were significantly lower in cluster 1 cells (proportion = 0.25, $p = 0.04$) compared to cells in cluster 2, 3 and 5. Also, when comparing the TGFB2 expression among TGFB[+] cells, we identified that TGFB2 expression was significantly higher in contractile VSMC compared to the other three VSMC clusters ($p = 2.0 \times 10^{-3}$; Supplementary Fig. 6). Additionally, we validated

our main findings of the four VSMC clusters, as well as the immune cell clusters by prediction and/or main marker genes expression using a publicly available single-cell RNA sequencing dataset of human carotid plaque cells (detailed description in "Methods"; Supplementary Figs. 7 and 8)[18]. Using this publicly available dataset,

we identified higher *TGFB2* expression in the proximal region (Supplementary Fig. 9a), in particular in contractile VSMC compared to other VSMC (Supplementary Fig. 9b), and confirmed that *TGFB2* and *TGFB3* are most prominently expressed by non-immune cells (Supplementary Fig. 10).

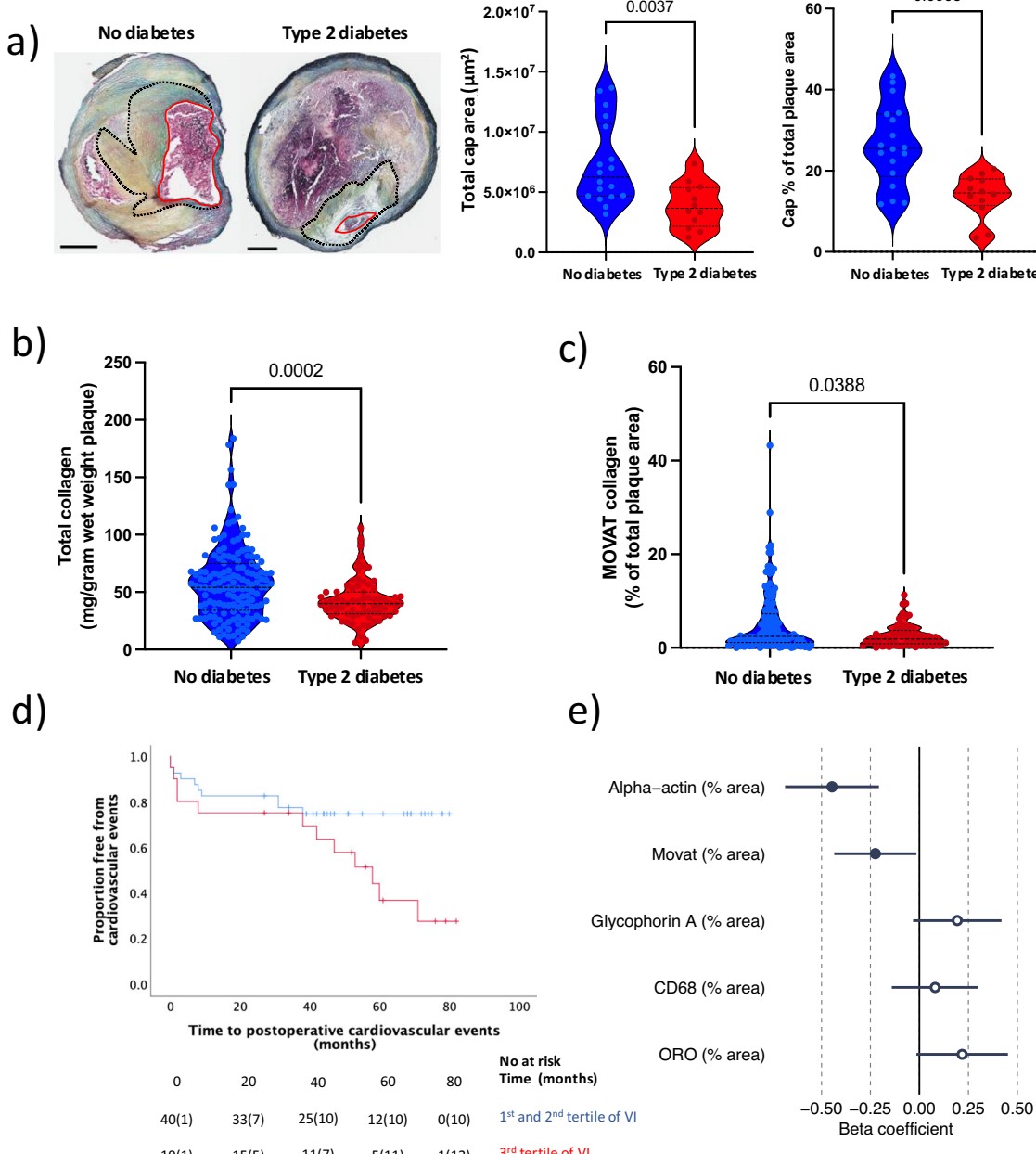

**Fig. 1 | Less collagen and vascular smooth muscle cells are associated with future cardiovascular events in type 2 diabetes . a** Atherosclerotic plaques from patients with type 2 diabetes (T2D; *n* = 12 patient samples) have smaller fibrous caps (cap area and % of plaque area) compared to plaques from patients without diabetes (*n* = 18 patient samples). MOVAT pentachrome staining was used to assess collagen. Dark dotted line marks the fibrous cap. Red line marks lumen. Scale bars 1 mm. T2D plaques (*n* = 72 patient samples) have significantly lower levels of collagen compared to plaques from patients without diabetes (*n* = 147 patient samples) measured in **b** whole plaque tissue homogenates and **c** histologically in the most stenotic region of the plaque (T2D, *n* = 68; no diabetes, *n* = 135 patient samples). Data is presented as violin plots (lines indicating median and interquartile range). Two-tailed Mann–Whitney *U*-test was used to determine the level of significance. **d** Kaplan–Meier curve showing an increased risk for cardiovascular

events among T2D patients with a calculated plaque vulnerability index (VI) in the 3rd tertile (red line) compared to patients with a plaque VI in the 1st and 2nd tertiles (blue line; *p* = 0.03). Numbers of events are shown in brackets. Two-sided log rank test was used (*n* = 61 patient samples). **e** A multiple linear regression analysis identified (ordered by absolute value of beta) smaller plaque areas (% plaque area) of collagen (MOVAT) and vascular smooth muscle cells (alpha-actin) as the histological components with the strongest influence on the calculated VI in plaques from T2D patients (*n* = 61). Beta coefficients and their 95% confidence interval are shown. *p* = 0.001 (alpha-actin), *p* = 0.04 (Movat), *p* = 0.10 (Glycophorin A), *p* = 0.48 (CD68) and *p* = 0.07 (ORO), according to the two-sided Student's *t*-tests. Solid points depict significant associations (*p* < 0.05). T2D type 2 diabetes, ORO Oil Red O, VI vulnerability index. Source data are provided in a source data file.

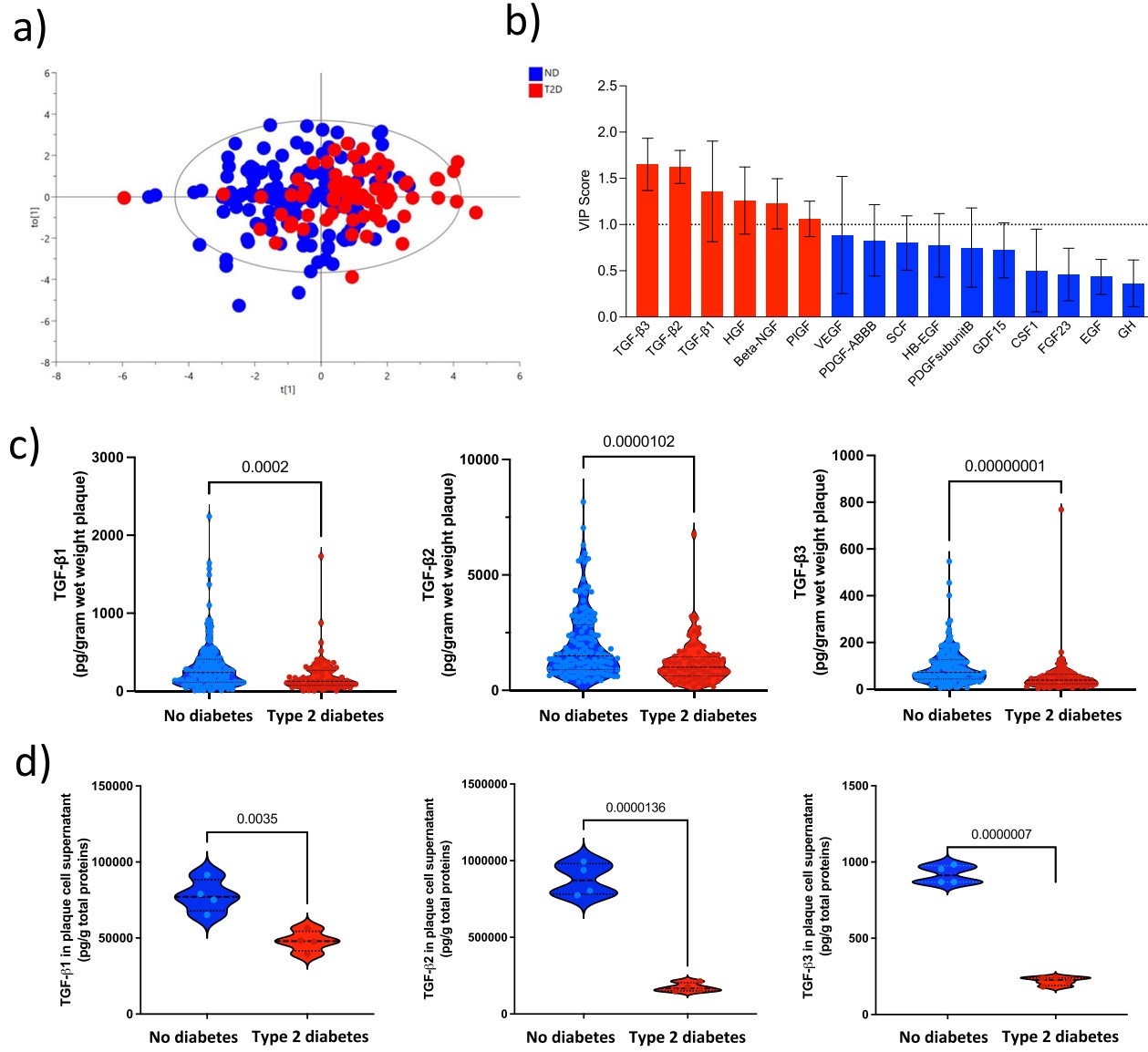

**Fig. 2 | TGF-β2 levels separated plaques from patients with T2D from plaques from patients without diabetes. a** Orthogonal partial least squares discriminant analysis (OPLS-DA) of 16 different growth factors measured in atherosclerotic plaques demonstrated a clear separation of plaques from patients with type 2 diabetes (T2D; red) and without diabetes (blue) (R²Y (cum) = 0.19, Q² (cum) = 0.14; CV-ANOVA *p*-value = 1.1 × 10⁻⁶; *n* = 218 patient samples). **b** TGF-β1, -β2 and -β3 were identified as the three top proteins separating T2D plaques from plaques from patients without diabetes based on their variable influence on projection (VIP) scores from OPLS-DA (*n* = 218 patient samples). Data presented as a bar graph of VIP scores and 95% confidence intervals. Growth factors with VIP scores greater than 1 are highlighted in red. **c** Total protein levels of TGF-β1, -β2 and -β3 were reduced in T2D plaque tissue homogenates (*n* = 72 patient samples) compared to plaque tissue homogenates from patients without diabetes (*n* = 147 patient samples). Data presented as violin plots with lines representing median and interquartile range

(IQR; 25th to the 75th percentile). Two-sided Mann–Whitney *U*-tests were used. **d** Plaque cells from donors with T2D released significantly lower levels of TGF-β isoforms compared to cells from patients without diabetes (*n* = 4 biological replicates). Live plaque cells were isolated from human carotid plaques and maintained in vitro for 17 h before TGF-β protein levels were assessed in cell culture supernatant. All values were normalized against total proteins. Data is presented as violin plots where lines represent median and interquartile range (IQR; 25th to the 75th percentile). Two-sided Mann–Whitney *U*-tests were used. TGF-β transforming growth factor-β, HGF hepatocyte growth factor, Beta-NGF beta-nerve growth factor, PIGF placental growth factor, VEGF vascular endothelial growth factor, PDGF platelet-derived growth factor, SCF stem cell factor, HB-EGF, heparin-binding EGF like growth factor, GDF growth differentiation factor, CSF-1 colony-stimulating factor-1, FGF fibroblast growth factor, EGF epidermal growth factor, GH growth hormone. Source data are provided in a source data file.

## VSMC differentiation induces TGFB2 expression

To investigate if the differentiation of synthetic/proliferative VSMCs into a contractile state would affect *TGFB2* expression, synthetic/proliferative human coronary arterial smooth muscle cells (HCASMC) were differentiated using VSMC differentiation supplement. Importantly and in line with our ScRNAseq, the expression of *TGFB2* (Fig. 3e) increased together with contractile cell phenotype markers (Fig. 3f) upon differentiation.

On a protein level, plaque TGF-β2 levels correlated to alpha-smooth muscle actin⁺ plaque area in both T2D and ND plaques (*r* = 0.29, *p* = 0.01 and *r* = 0.18, *p* = 0.03, respectively; Supplementary Fig. 11a, b). Using bulk RNA sequencing (RNAseq) on 22 plaques obtained from patients with T2D, we confirmed that *TGFB2* showed stronger correlations to genes representing contractile VSMCs than synthetic VSMCs (Supplementary Fig. 11c). Taken together, contractile VSMC (*ACTA2*ʰⁱᵍʰ) appeared to be the foremost source of

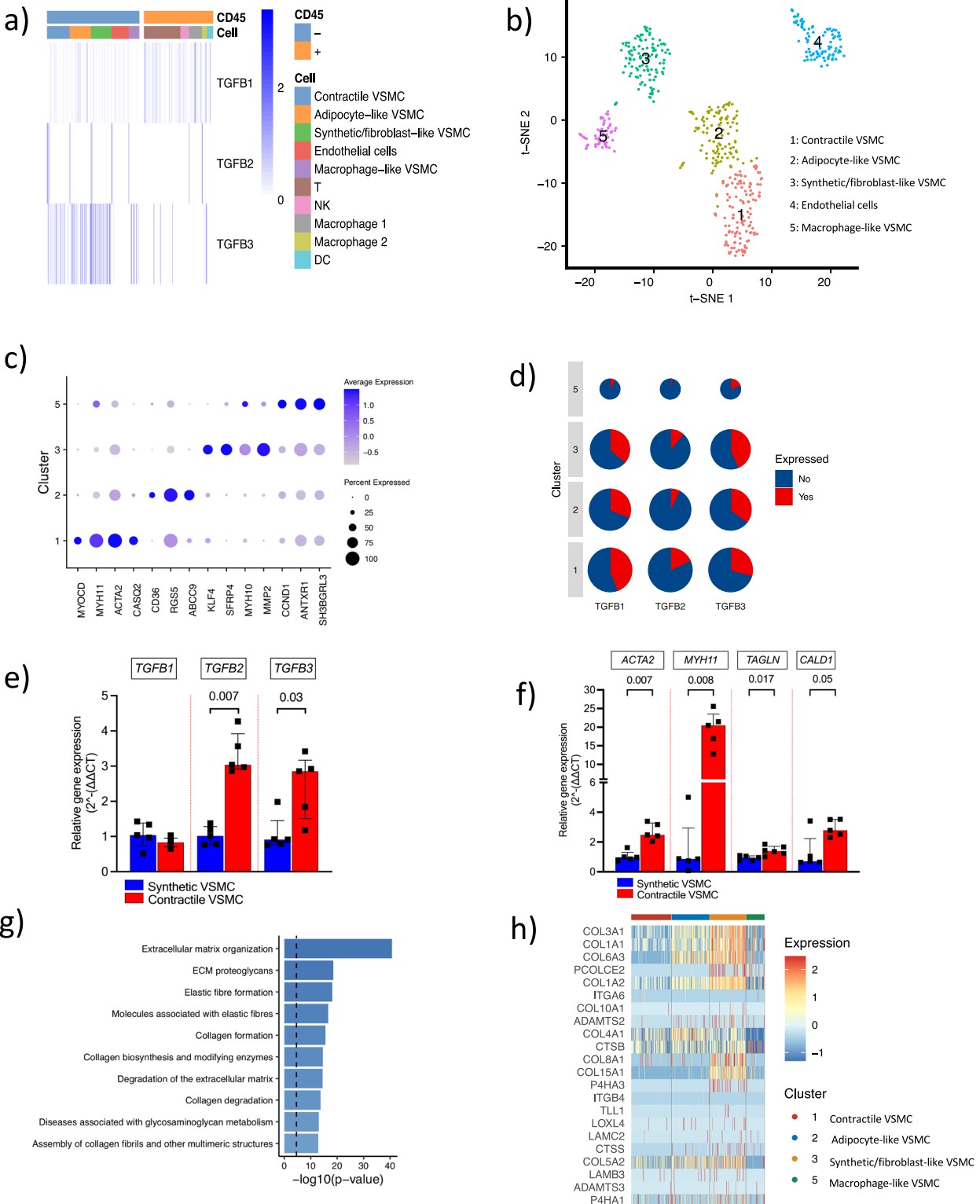

TGF-ß2 in human plaques, and *TGFB2* was also upregulated upon differentiation.

**Fibrous repair is associated with synthetic VSMC**

Next, we explored which VSMC phenotypes that were associated to fibrous tissue repair. Pathway analyses showed that the synthetic/fibroblast-like VSMCs (cluster 3) were enriched in fibrous repair-associated pathways (including extracellular matrix organization ($p = 1.2 \times 10^{-34}$), collagen formation ($p = 2.0 \times 10^{-13}$) and collagen biosynthesis ($p = 8.3 \times 10^{-13}$); Supplementary Fig. 12 and Fig. 3g). Synthetic/fibroblast-like VSMC (cluster 3) also expressed higher levels of *COL1A1*, *COL4A1* and *COL3A1* (based on collagen formation pathway; Fig. 3h). This suggests that synthetic/fibroblast-like VSMCs are key for the fibrous repair process.

**Fig. 3 | Four phenotypes of vascular smooth muscle cells are present in the human atherosclerotic plaque and contractile vascular smooth muscle cells are the major source of *TGFB2*. a** Human atherosclerotic carotid plaque CD45⁺ and CD45⁻ cell phenotypes identified by annotation of single-cell RNA sequencing analysis and their expression of the three *TGFB* isoforms (*n* = 855 cells). The color scale represents log-normalized counts of gene expression. **b** t-SNE plot visualizing the 5 major CD45⁻ plaque cell clusters. Clusters 1: contractile smooth muscle cells, Cluster 2: adipocyte-like smooth muscle cells, Cluster 3: synthetic/fibroblast-like smooth muscle cells, Cluster 4: endothelial cells, Cluster 5: macrophage-like smooth muscle cells. **c** A dot plot showing expression of key cell phenotype markers and vascular smooth muscle cell (VSMC) transcription factors in cluster 1, 2, 3 and 5. The dot size represents the percentage of cells expressing a given gene in the respective cluster, and the color scale represents the scaled gene expression. **d** Pie charts showing the percentage of cells in the four major smooth muscle cell clusters expressing the *TGFB1*, *TGFB2* and *TGFB3* isoforms. Pie size denotes the number of cells for each cluster. **e** Gene expression levels of *TGFB2* and **f** contractile VSMC markers were induced upon differentiation of human coronary arterial smooth muscle cells in smooth muscle cells differentiation supplement. For (**e**) and (**f**), boxes mark the median levels and interquartile range (25th to the 75th percentile), *n* = 5 biological replicates in each group. Two-sided Mann–Whitney *U*-tests were used for group comparisons. **g** Overexpressed genes in the cluster 3 smooth muscle cells, identified as synthetic/fibroblast-like smooth muscle cells, were enriched in pathways associated with fibrous repair. Vertical dashed line corresponds to a Bonferroni-corrected *p*-value of 0.05. **h** Heatmaps showing expression of the most variable genes in the "Collagen formation" pathway in the four smooth muscle cell clusters. The color scale represents the scaled gene expression. Source data are provided in the source data file.

## TGF-ß2 induces collagen gene expression

Among the three major ECM components of the human plaques (elastin, glycosaminoglycans and collagen), collagens are the most important structural proteins in the formation of the stabilizing cap. Hierarchical clustering on total plaque levels of the measured growth factors and collagen showed that three TGF-β isoforms and plaque collagen were grouped together, suggesting close relationships shared between them (Fig. 4a). Further correlation analyses demonstrated that the total plaque levels of free TGF-β2 protein correlated to total plaque levels of collagens in both ND plaques and T2D plaques ($r = 0.24$, $p = 0.003$ and $r = 0.41$, $p = 4.0 \times 10^{-4}$ respectively; Fig. 4b).

Intrigued by these findings, we investigated if TGF-β2 stimulation modulates collagen gene expression in vitro. The potential effects of TGF-β1 stimulation have been well studied, but to our knowledge, no such studies have been conducted regarding the TGF-β2 stimuli on VSMCs. In order to assess the effect of TGF-β2 on collagen formation, synthetic/proliferative HCASMC were used. First, we concluded that stimulation of synthetic/proliferative HCASMC with increasing concentration of TGF-βs resulted in increased phosphorylation of SMAD2 (Ser467) and SMAD3 (Ser423/425; Fig. 4c, d and Supplementary Fig. 13). Second, we identified that the TGF-β2 induced SMAD phosphorylation resulted in increased mRNA levels of the collagen genes: *COL1A1*, *COL3A1* and *COL4A1* ($p = 4.1 \times 10^{-5}$, $1.2 \times 10^{-4}$ and $4.1 \times 10^{-5}$ respectively; Fig. 4e). This indicates that an interaction between the fibrous repair associated synthetic/fibroblast-like cells and the TGF-β2 expressing contractile VSMCs is needed for an efficient fibrous tissue repair response. To elucidate this hypothesis, synthetic/proliferative HCASMC were incubated with supernatants obtained from cultured differentiated contractile HCASMC, which confirmed an increased collagen gene expression (Fig. 4f).

## MMP2 activity is lower in T2D plaque cells

Reduced T2D plaque levels of all three TGF-β isoforms could be either due to an impaired enzymatic cleavage of TGF-β-LAPs complexes or less TGF-β expressing/secreting cells. Considering that plaque levels of the three TGF-β isoforms correlated strongly with each other ($r = 0.56$–$0.67$, Supplementary Fig. 14) and that the cell phenotypes expressing the three isoforms are different, this suggests that even the extracellular TGF-β cleavage is affected in T2D. MMP2 and -9 have been shown in vitro to be key enzymes in TGF-β-LAPs complex cleavage[18]. Furthermore, MMP2 is crucial for the migration of smooth muscle cells from the media into the intima in the vascular wall (summarized in Fig. 5a)[19,20]. Therefore, we compared plaque levels of total MMP2 and -9 and confirmed, in a larger cohort, our previous finding that MMP2 levels are significantly lower in T2D, plaques, whereas no difference in MMP9 levels was detected (49,255 (IQR 29,302–72,040) vs. 68,905 (IQR 40,639–146,320) pg/gram wet weight plaque, $p = 5.0 \times 10^{-4}$ Fig. 5b)[9]. Moreover, MMP2 correlated to free plaque levels of TGF-β2 among ND patients but not among T2D patients ($r = 0.224$, $p = 0.009$ and $r = 0.018$, $p = 0.88$, Fig. 5c). The lack of correlation in T2D plaques further supports the hypothesis of reduced MMP2 activity in T2D. In contrast to MMP2, MMP9 did not correlate to free TGF-β2 levels in either of the two groups.

Finally, to investigate if MMP2 activity is affected in T2D plaques, live plaque cells were isolated from human plaques and MMP2 activity was measured after 24 h of in vitro culture without further stimuli. MMP2 activity was reduced in T2D cell cultures (0.84 SD 0.03 fold change active/pro-MMP2 vs. 1.1 SD 0.06 fold change active/pro-MMP2, $p = 0.003$; Fig. 5d). Interestingly, the ScRNAseq analysis, revealed that synthetic/fibroblast-like VSMCs (cluster 3) was the main source of *MMP2* in plaques, whereas *MMP9* was only expressed by small proportion of VSMCs (cluster 2,5 and 4) as well as macrophages (Fig. 5e and Supplementary Fig. 14b).

## Hyperglycemia reduces MMP2 activity

Using multiple linear regression, we next assessed clinical factors potentially affecting free TGF-β2 levels. Interestingly, hemoglobin A1c (HbA1c) was inversely associated with free TGF-β2 levels (Supplementary Table 6), suggesting that hyperglycemia may affect MMP2 activity. As synthetic/fibroblast-like VSMCs were identified as the main cellular plaque source of MMP2, synthetic/proliferative HCASMC were stimulated with high and normoglycemic glucose concentrations. Interestingly, MMP2 activity was found to be reduced at high glucose concentration (0. 69 SD 0.21 vs 1 SD 0.30 -fold change active/pro MMP2, $p = 0.038$; Fig. 5f). Likewise, HbA1c levels correlated inversely to plaque levels of free TGF-β2 ($r = -0.27$, $p = 0.001$) as well as to plaque collagen content ($r = -0.19$, $p = 0.03$; Supplementary Fig. 15) and collagen plaque areas ($r = -0.19$, $p = 0.03$; Supplementary Fig. 15). Importantly, a strong inverse correlation was identified between HbA1c and % cap area ($r = -0.61$, $p = 0.004$; Supplementary Fig. 15).

MMP2 plays a key role in VSMC migration[19,20], and a reduced MMP2 activity may then affect the migration of VSMCs into the cap. Therefore, a wound scratch assay was used to assess synthetic/proliferative HCASMC migration in normo- and hyperglycemic conditions. In further support for the hyperglycemic effect on MMP2 activity, hyperglycemia reduced HCASMC migration (8 h 34.89 SD 6.4 vs 44.34 SD 9.3% wound closure $p = 0.06$; 12 h 45.96 SD 2.2 vs 57.14 SD 9.2% wound closure $p = 0.01$; 16 h 63.7 SD 5.991 vs 72.02 SD 4.4% wound closure $p = 0.01$, respectively; Supplementary Fig. 16). Taken together, these results imply that poor glycemic control could affect fibrous repair in T2D by modulating MMP2 activity, which in turn could reduce TGF-β2 cleavage, VSMC migration and collagen synthesis.

## VSMC interaction and differentiation are affected in T2D

To locate VSMC phenotypes in the plaque and to explore the interplay between VSMC in the fibrous cap (where fibrous repair occurs; summarized in Fig. 6a), as well as to validate the lower TGF-β2 expression in T2D plaques, a deep spatial transcriptomics was performed on a validation cohort of nine human carotid plaques. The clinical characteristics of the validation cohort are provided in Supplementary Table 7.

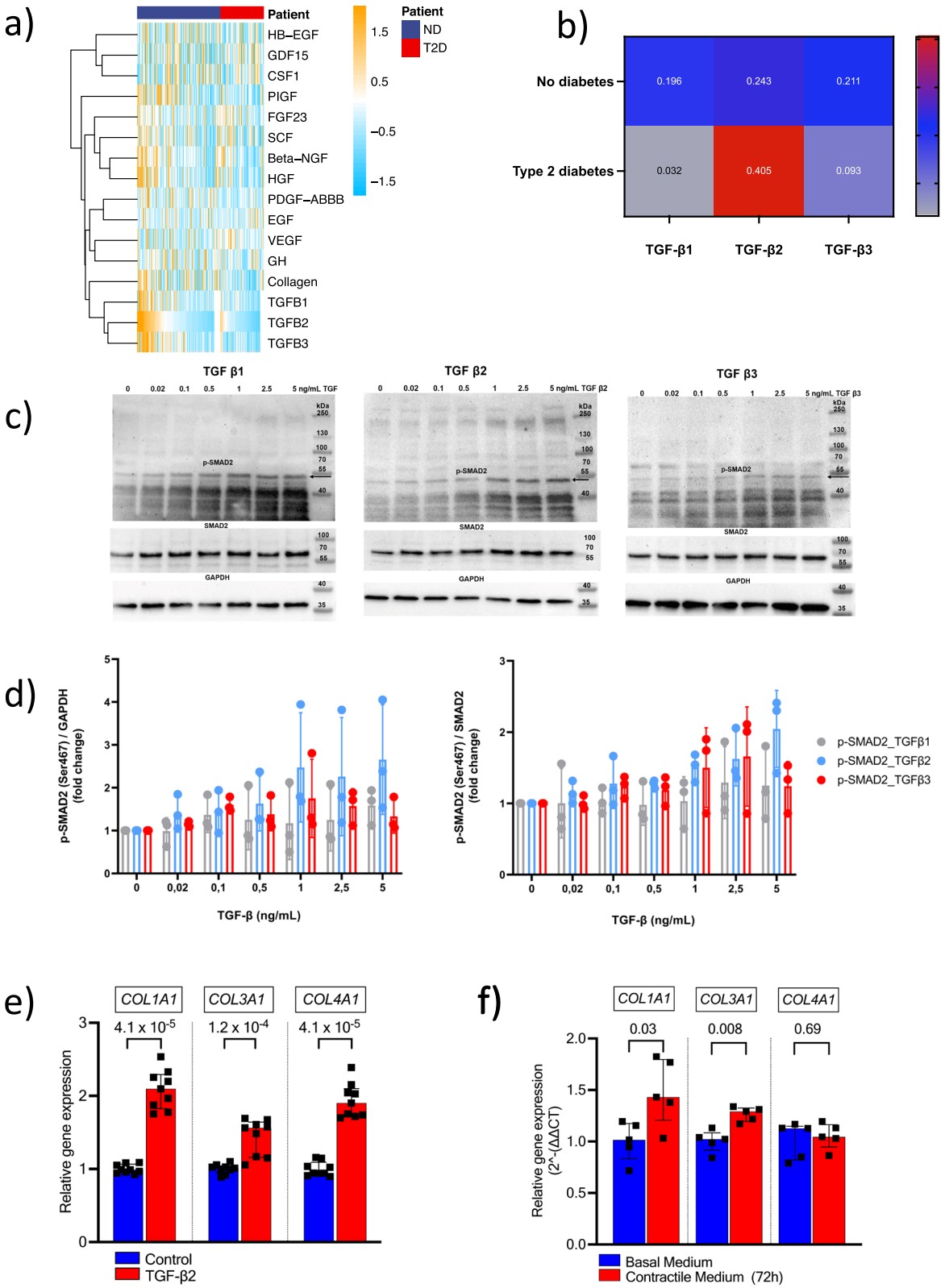

Importantly, when analyzing all *PTPRC⁻* spots, we identified that the CD45⁻ cluster 1 cells (contractile VSMC) were enriched in the cap region, whereas cluster 3 cells (synthetic/fibroblast-like VSMCs) were enriched in the interface to the media. Cluster 2 and 5 cells (adipocyte- and macrophage-like VSMCs) showed a more widespread distribution in the plaque (Fig. 6b). Macrophages were located close to the core, whereas T-cells were broadly distributed (Supplementary

Fig. 17). In line with the single cell clustering, *TGFB2* expression was detected in the cap region of plaque in areas identified as cluster 1 VSMCs (contractile phenotype; Fig. 6b). Interestingly, even though the majority of VSMCs in the cap were identified as contractile VSMCs, infiltrating synthetic/fibroblast-like VSMCs were detected in close proximity, again suggesting an interaction between the two cell phenotypes in the fibrous cap.

**Fig. 4 | TGF-β2 shows strong associations with collagen. a** Among all growth factors measured in the plaque tissue, only the three TGF-β isoforms grouped together with plaque collagen. Hierarchical clustering of variables was done using Euclidean distance metric, where data for each variable (rows) was standardized. The color key on the right indicates the increased standardized value of each variable in the plaques. Patient samples are grouped into type 2 diabetes (T2D; red) and no diabetes (ND; blue). $n = 218$ patient samples. **b** Heatmap showing Spearman's correlation coefficients between the three TGF-β isoforms and collagen levels in plaques from patients without diabetes ($n = 147$ patient samples) and patients with T2D ($n = 72$ patient samples). The color scale represents the correlation coefficient. **c, d** TGF-β1, -β2 and -β3 induced SMAD2 phosphorylation in synthetic human coronary arterial smooth muscle cells (HCASMC). HCASMC were treated for 15 min with up to 5 ng/mL of TGF-β1, - β2, or- β3. Proteins were extracted by direct lysis in loading buffer, separated by 10% SDS-PAGE and subjected to Western blotting detecting phospho-SMAD2 (Ser467) and GAPDH (left panel) or total SMAD2 (right panel) as loading controls. Values are expressed as means ± standard deviation and are normalized to untreated controls. $n = 3$ biological replicates from three independent experiments. **e** TGF-β2 stimuli (5 ng/mL, 72 h) induced gene expression levels of *COL1A1*, *COL3A1* and *COL4A1* in HCASMC. Boxes and bars mark the median and the interquartile range, $n = 9$ biological replicates in each group from three independent experiments. Two-sided Mann–Whitney *U*-tests were used. **f** Culturing synthetic/proliferative HCASMC in cell culture supernatants obtained from differentiated contractile HCASMC increased gene expression levels of *COL1A1* and *COL3A1*. Boxes and bars indicate the median and the interquartile range, $n = 5$ biological replicates in each group. Two-sided Mann–Whitney *U*-tests were used. TGF-β transforming growth factor-β, HGF hepatocyte growth factor, Beta-NGF beta-nerve growth factor, PlGF placental growth factor, VEGF vascular endothelial growth factor, PDGF platelet-derived growth factor, SCF stem cell factor, HB-EGF heparin-binding EGF like growth factor, GDF growth differentiation factor, CSF-1 colony-stimulating factor-1, FGF fibroblast growth factor, EGF epidermal growth factor, GH growth hormone. Source data are provided in the source data file.

When comparing the prediction scores of cluster 3 cells in the plaque cap regions between T2D and ND, significantly greater scores were found in T2D (Fig. 6c), suggesting a large number of the cluster 3 cells in T2D caps compared to ND caps. By contrast, the prediction scores of cluster 1 cells were significantly lower in T2D caps (Fig. 6c). The lower number of predicted contractile (*ACTA2*<sup>high</sup>) VSMC spots in T2D caps was confirmed by immunohistochemistry showing a reduced total number of smooth muscle alpha-actin⁺ cells as well as lower percentage of smooth muscle alpha-actin⁺ cells (% of all cells) in T2D plaque caps (115 (IQR 70–169) vs 620 (84–884) cells, $p = 0.013$ and 7.8 (IQR 4.5–17.3) vs 16 (IQR 7.9–28.6)% of all cells, $p = 0.04$; Supplementary Fig. 18). In line with the plaque TGF-β2 protein levels and lower frequency of predicted contractile VSMC in T2D caps, caps from T2D plaques showed a larger proportion of *TGFB2*⁻ spots compared to *TGFB2*⁺ spots ($p = 0.03$; Fig. 6d).

This opposite pattern of synthetic and contractile VSMC phenotypes in T2D caps is contradictory to the reduced migration of synthetic VSMCs into the cap. Yet, considering the lower number of contractile VSMC in T2D caps, this could be due to a limited differentiation of synthetic VSMC into a contractile TGFB2⁺ transcriptional state.

TGF-β1 is an important inducer of cell differentiation but it remains unknown, to our knowledge, if TGF-β2 can induce VMSC differentiation. To test this hypothesis, synthetic/proliferative HCASMC were exposed to TGF-β2 in vitro. Importantly, TGF-β2 stimuli clearly induced the expression of contractile cell markers, including *TAGLN*, *CALD1* and *CNN1* (Fig. 6e), showing that TGF-β2 in itself plays an important role in the VSMC differentiation.

In summary, this shows that the majority of VSMCs present in the cap are in a contractile state and represent the main source of *TGFB2* expression. Although the migration of synthetic VSMCs is reduced by hyperglycemia in vitro, no difference was seen in the total number of synthetic VSMCs in the caps. Instead, reduced numbers of contractile VSMCs were detected in the cap, potentially due to reduced VSMC differentiation induced by TGF-β2. In the end, this may lead to a negative feedback loop with less *TGFB2*-expressing cells, less secreted TGF-β2 and an impaired collagen synthesis and cap formation.

### Associations between TGF-β2 and clinical characteristics

In plaques obtained from patients without diabetes, TGF-β2 correlated inversely with patient age ($r = -0.31$, $p = 1.0 \times 10^{-4}$) and positively to estimated glomeruli filtration rate (mL/min; $r = 0.25$, $p = 0.002$), whereas no significant correlations between clinical characteristics and plaque TGF-β2 were observed for patients with T2D. TGF-β2 levels were significantly higher in females compared to males, both among individuals with T2D (1216 (IQR 859–1699) pg/gram wet weight plaque vs. 858 (IQR 500–1388) pg/gram wet weight plaque; $p = 0.04$) and without diabetes (2552 (IQR 1106–4248) pg/gram wet weight plaque vs.

1363 (IQR 839–2228) pg/gram wet weight plaque; $p = 0.003$). No significant associations between plaque levels of TGF-β2 and hypertension, smoking, statin treatment, diabetes treatment (insulin/metformin/food control only or the combination of insulin and metformin), nor usage of RAAS inhibitors or beta-blockers were detected among patients without diabetes or with T2D.

## Discussion

Here, we show that T2D plaques have smaller fibrous caps due to hyperglycemia-dependent reduction of MMP2 activity, affecting synthetic vascular smooth muscle migration and TGF-β2 cleavage, which in turn limits VSMC differentiation and collagen formation. These molecular mechanisms are embodied by close interactions between *TGFB2*⁺ contractile and *MMP2*⁺ synthetic VSMCs within the fibrous cap. Furthermore, the impaired fibrous tissue repair and reduction of fibrous cap thickening in T2D is associated with a higher risk for future CV events and poorer glycemic control.

Even though the incidence of CV events has declined over the past 15 years, due to improved medical treatments, T2D remains one of the major risk factors[1]. This implies that other molecular mechanisms need to be targeted in T2D. Inefficient resolution of inflammation and fibrous repair, counterbalancing processes ongoing in parallel with inflammatory pathways, have been put forward as such mechanisms[21]. Clinical trials using intravascular ultrasound (IVUS) and ultrasound have shown that plaques in both coronary and carotid arteries of T2D patients are more echolucent and have thinner fibrous caps, reflecting a vulnerable plaque phenotype[11–14,22,23]. Moreover, both echolucent plaques and plaques with a thin fibrous cap have been shown to be at higher risk to rupture, indicating the clinical importance of the fibrous repair process[23,24]. The present study identified that T2D plaques have smaller fibrous caps, lower total collagen levels and smaller total plaque areas of collagen. Next, using the previously published histological ratio (VI) reflecting the balance between vulnerable plaque components (neutral lipids, CD68⁺ cells and hemorrhage) and fibrous repair-associated plaque components (collagen and alpha-actin⁺ cells)[15,16] we showed that T2D individuals with high plaque vulnerability index were at significantly higher risk for future CV events. Moreover, collagen and alpha-actin⁺ plaque area (VSMC) were the components with the greatest influence on the plaque vulnerability index in T2D subjects only, again highlighting the role of compromised fibrous repair processes in determining plaque vulnerability in T2D subjects.

As growth factors are key components in both fibrous repair responses and atherosclerosis, known to mediate cellular transcriptional changes in both macrophages and smooth muscle cells, we used an unbiased approach combining major growth factors to identify the most important ones discriminating T2D plaques. Interestingly, the three isoforms of TGF-β were identified to have the greatest effect on

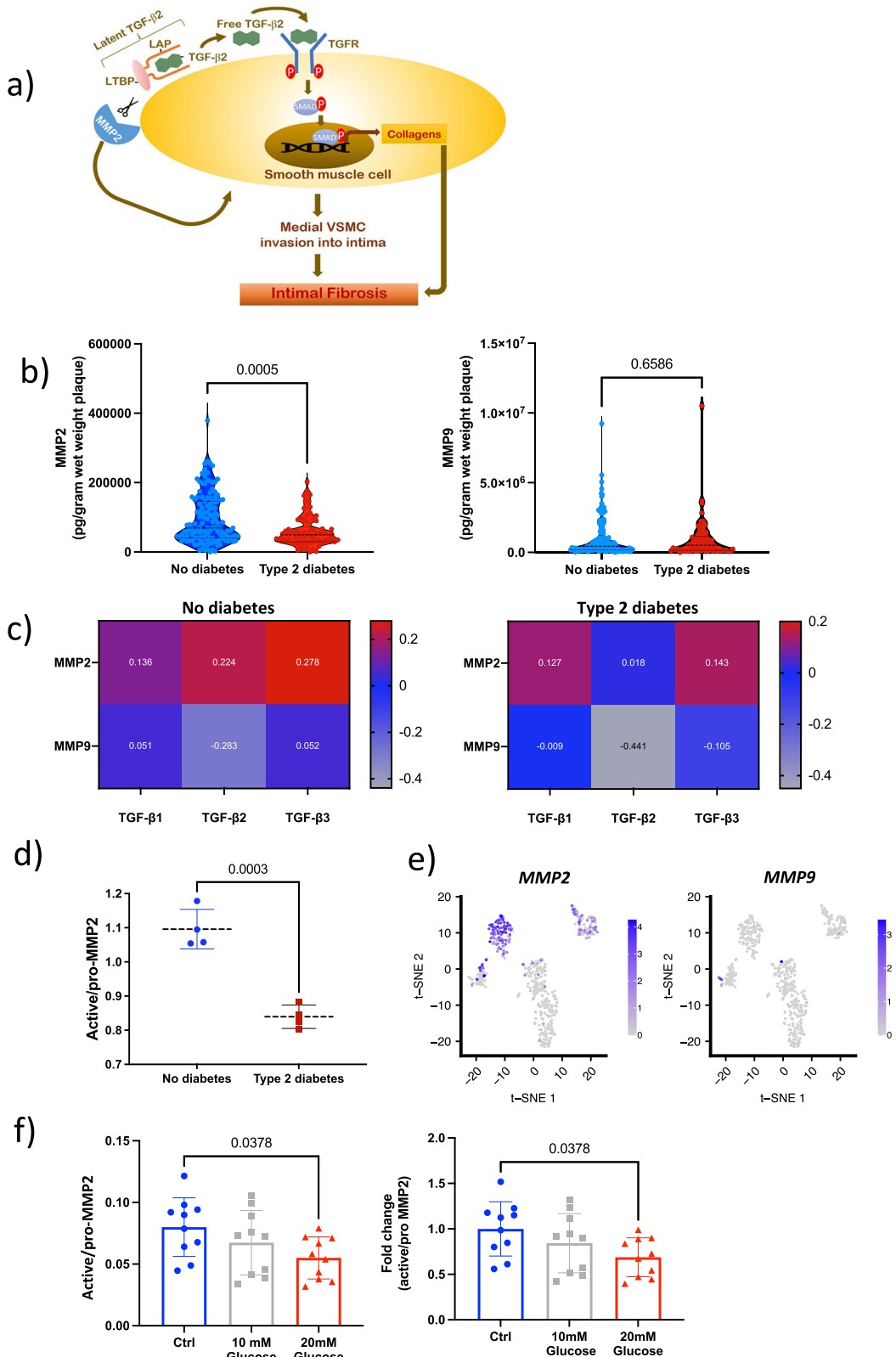

separating T2D plaques from ND plaques. The three TGF-β isoforms are multifunctional cytokines with a highly similar peptide structure which exerts a wide range of effects on both fibrous repair and immunological responses through TGF-β receptor activation of the canonical SMAD pathway[25,26]. Among the three, most atherosclerotic studies have focused on TGF-β1[27,28] Nevertheless, TGF-β2 has been shown to limit inflammatory responses in non-vascular and vascular

diseases by dampening macrophage activation and concurrent chemokines[17,29]. Studies on retinal epithelial cells and astrocytes have also suggested that TGF-β2 may stimulate collagen synthesis[30,31]. However, if TGF-β isoforms contribute to fibrous tissue repair responses in human atherosclerosis and if TGF-β levels are affected by T2D have so far not been studied. Among the three isoforms, TGF-β2 was the most abundant in both plaques from patients with and without

**Fig. 5 | MMP2 is released from synthetic vascular smooth muscle cells and MMP2 induced TGF-β2 cleavage is reduced by hyperglycemia. a** Schematic figure illustrating the interplay between matrix metalloproteinase 2 (MMP2) and TGF-LAP complex cleavage to induce collagen production through the release of TGF-β2. **b** MMP2 but not MMP9 levels in plaque tissue homogenates were significantly lower in plaques from patients with type 2 diabetes (T2D; $n$ = 70 patient samples) compared to patients without diabetes ($n$ = 134 patient samples). Results are presented as violin plots with dotted lines indicating the median levels and the interquartile range (25th to the 75th percentile). Two-sided Mann–Whitney $U$-tests were used. **c** Heatmap showing Spearman correlation coefficients between the three TGF-β isoforms and MMP2 and MMP9 levels in plaques from patients without diabetes ($n$ = 134 patient samples) and patients with T2D ($n$ = 70 patient samples). The color scale represents the correlation coefficient. **d** Zymography showing reduced MMP2 activity in human atherosclerotic plaque cell isolates from patients with T2D compared to cells from patients without diabetes. $n$ = 4 biological replicates in each group. Lines indicating mean (dashed) ± standard deviation (SD) of the ratio between active and pro-MMP2. Two-sided Student's $t$-test was used. **e** t-SNE plot demonstrating that *MMP2* is mainly expressed by synthetic vascular smooth muscle cells (cluster 3 cells) in the human plaque single-cell RNA sequencing. The color scale represents log-normalized counts of gene expression. **f** Hyperglycemic conditions reduced MMP2 activity, assessed by zymography, in synthetic human coronary arterial smooth muscle cells (HCASMC). $n$ = 10 biological replicates from five independent experiments. One-way repeated measure ANOVA followed by Dunnett post-hoc tests (two-sided) were used. Results are presented as mean ± SD. TGF transforming growth factor, MMP matrix metalloproteinase. Source data are provided in the source data file. Credit: Part **a** created with http://BioRender.com. Released under a Creative Commons Attribution Non-Commercial NoDerivatives 4.0 International License (CC BY NC ND 4.0).

T2D and the only isoform that was negatively associated with plaque vulnerability index. Taken together, this provides evidence for a major role of TGF-β2 in human arterial fibrous tissue repair.

The human plaque CD45[+] cell architecture has been well described, whereas the cellular landscape of CD45[−] cells remains to be fully characterized, and no previous studies have focused on the fibrous repair responses[32–34]. A few recent studies have provided important evidence for the heterogeneity of VSMC phenotypes comparing different arterial sites as well as cells obtained from human and mouse plaques[35–37]. However, our CD45[−] cell characterization shows an overview of VSMC phenotypes and where these cells are spatially located. Using deep ScRNAseq of isolated human plaque cells, we identified 5 distinct clusters of CD45[−] cells: four cluster of VSMCs (contractile, synthetic/fibroblast-like, adipocyte-like and macrophage-like) and one cluster of endothelial cells. The contractile and synthetic VSMC were separated by phenotype markers and transcription factors (*KLF*4 and *MYOCD*). *MYOCD* is the key activator, whereas *KLF*4 is a well-described suppressor of contractile genes in VSMCs[38]. In support of their role in human plaque VSMC differentiation, *MYOCD* was significantly increased in contractile VSMCs and *KLF*4 in synthetic VSMCs. Furthermore, our pathway analyses showed that synthetic/fibroblast-like VSMCs were enriched in fibrous repair pathways including "Collagen formation" and "Extracellular matrix organization", indicating their importance in the fibrous repair responses. However, *TGFB2* expression was significantly higher among contractile VSMCs compared to other VSMC clusters, suggesting that the contractile VSMCs are needed to stimulate the fibrous repair responses.

By spatial deep sequencing, we provide a micro-architectural map of VSMC cell phenotypes based on their transcriptional state in the human plaque. Synthetic VSMC were generally seen in the interface to the media, whereas adipocyte-like VSMCs were more generally distributed in plaque, including the core regions. In opposite to the other VSMC clusters, contractile (*ACTA2*[high]) VSMC clusters were predominantly located to the fibrous cap, a region known to be enriched in alpha-actin expressing cells. Contractile VSMCs in the cap likely originate from migrating intimal smooth muscle cells, prone to display a synthetic transcriptional state upon activation by external stimuli[39,40]. The differentiation of migrating synthetic VSMC back to a contractile state has been considered to reflect an end stage of vascular repair responses[41]. Yet, the present study shows that the contractile VSMC likely play an active important role in the actual fibrous repair process. *ACTA2*[high] contractile VSCM clusters in the fibrous cap were also *TGFB2*[+], supporting the ScRNAseq identified *TGFB2* overexpression among contractile VSMC. The increased expression of *TGFB2* was also confirmed in vitro in differentiated contractile HCASMC compared to synthetic/proliferative HCASMC. Importantly, in close proximity to the contractile VSMC clusters, infiltrating synthetic clusters were identified, indicating an interplay between the TGF-β2 releasing contractile VSMCs and the fibrous repair associated synthetic VSMCs in the fibrous cap formation.

TGF-β2 is known to induce collagen synthesis in other cells and diseases; however, its role in human arterial smooth muscle cells is seldom characterized. TGF-β2 induced rapid SMAD2/3 phosphorylation and increased the expression of collagen genes in human proliferative/synthetic HCASMC. In order to functionally validate the existence of this interaction, we stimulated synthetic HCASMC with cell culture supernatants from differentiated contractile HCASMC and detected an increase in collagen gene expression. Finally, of the three TGF-isoforms, TGF-β2 showed the strongest association with plaque collagen content. This suggests that contractile VSMCs also play a key role in the fibrous repair process by stimulating synthetic VSMC collagen synthesis.

Next, we intended to identify factors underlying a reduced fibrous repair ability in T2D plaques. Here we showed a dual effect of hyperglycemia reduced MMP2 activity on TGF-β2 levels and collagen in T2D plaques: affecting smooth muscle cell migration, TGF-β cleavage and thereby limiting VSMC differentiation. Not only were total MMP2 protein levels lower in T2D plaques, but also its activity. Total MMP2 also showed a positive association with TGF-β2 levels in ND plaques but not in T2D plaques, supporting an altered MMP2 activity specifically in T2D plaques. Previous studies investigating MMP2 activity in T2D are conflicting but there are studies showing both reduced MMP2 levels and activity in blood from T2D patients[42,43].

In the present study, HbA1c was the only factor associated with free TGF-β2 levels (as a measurement of MMP2 activity), suggesting that hyperglycemia may affect TGF-β2 cleavage. This finding is supported by a previous study showing a negative association between salivary MMP2 activity and HbA1c levels[44] and when exposing synthetic/proliferative HCASMC in vitro to high glucose, MMP2 activity was again reduced.

MMP2 is also a key contributor to VSMC migration by catalyzing the removal of the basal membrane[19,20], so we examined if migration of VSMC can be hindered by glucose. Interestingly, in line with decreased MMP2 activity, high glucose impaired the migration of synthetic/proliferative HCASMC. Nevertheless, the number of predicted synthetic VSMC cell spots (identified by spatial sequencing) was not reduced in T2D plaques. Instead, the number of predicted contractile VSMC spots and *TGFB2*[+] spots were lower in T2D caps. As TGF-βs affect the transcriptional state of several cell types[45], this imbalance between synthetic and contractile VSMC could be due to an impaired contractile *TGFB2*[+] cell differentiation caused by decreased MMP2 activity and concurrent TGF-β2 cleavage. Importantly, when stimulating synthetic HCASMC with TGF-β2, the gene expression of major contractile phenotype markers increased, indicating that TGF-β2 regulates contractile VSMC differentiation.

The clinical importance of the identified effect of hyperglycemia on fibrous repair could be questioned. Diabetes, prediabetes and hyperglycemia have repeatedly been associated with atherosclerotic complications, which have led to several large clinical prospective trials, including the UKPDS, VADT, ACCORD and ADVANCE exploring the effect of intensified glycemic control[46–49]. Surprisingly, none of

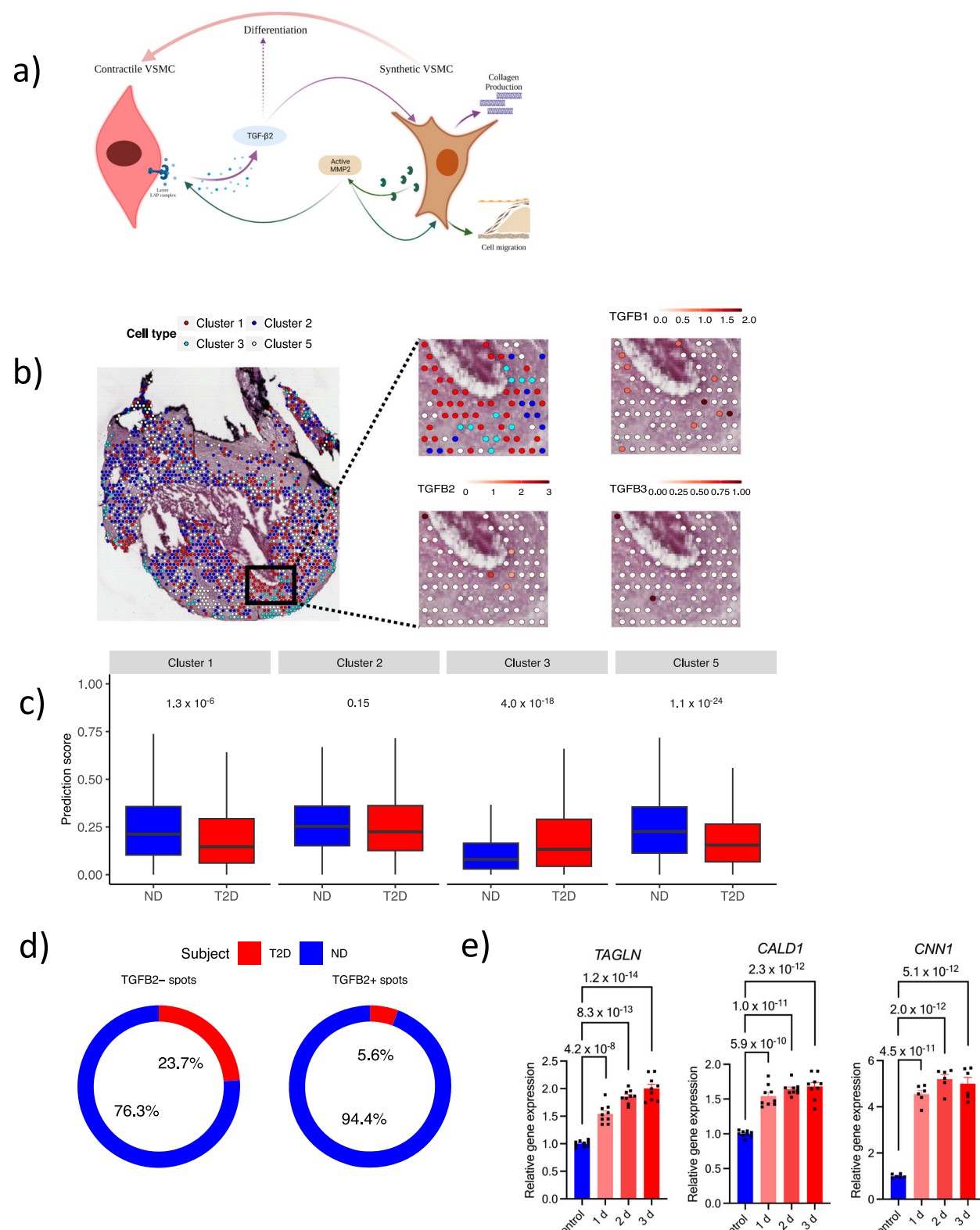

these studies were able to identify a significant effect on cardiovascular complications. However, both VADT and UKPDS showed positive effects of intensified glycemic control in their expanded follow-up studies[50,51], which is interesting from a biological perspective. Considering that plaque tissue median turnover has been shown to be approximately 10 years[52,53], trials focusing on atherosclerotic complications with a 5–6 years' follow-up are less likely to reflect major plaque composition changes. Together these findings suggest that

hyperglycemia may still have a role in atherosclerotic plaque phenotype changes. In further support for a potential role of hyperglycemia, HbA1c levels, as a marker of long-term hyperglycemia, were associated with lower levels of TGF-β2, plaque collagen and fibrous cap size.

There are limitations that need to be considered. First, the current study only examined atherosclerotic plaque biology in plaques from the carotid arteries. However, even if plaque morphology may differ between plaques in the coronary and the carotid arteries, key features

**Fig. 6 | Hyperglycemia reduces vascular smooth muscle cells migration and differentiation. a** Summary of TGF-β2 effects in the human plaque fibrous-repair process. **b** Visium spatial deep sequencing on human plaque tissue sections was used to characterize the spatial distribution of the four phenotypes of vascular smooth muscle cells identified by single-cell RNA sequencing in all *PTPRC* spots. Cluster 1 (contractile)-red. Cluster 2 (adipocyte-like)-dark blue. Cluster 3 (synthetic/fibroblast-like)- light blue. Cluster 5 (macrophage-like)- white. **c** The prediction scores to be contractile and macrophage-like vascular smooth muscle cells were lower in plaque caps from patients with type 2 diabetes (T2D, *n* = 4 patient samples) than patients without diabetes (*n* = 5 patient samples). Two-sided Student's *t*-tests were used. The boxes represent the interquartile range (IQR; 25th to the 75th percentile). The central line within each box marks the median. The whiskers extend from the minimum value, calculated as the lower quartile minus 1.5 times

the IQR, to the maximum value, determined as the upper quartile plus 1.5 times the IQR. **d** Proportions of *TGFB2*⁻ and *TGFB2*⁺ spots in plaque caps in relation to patients with or without T2D. **e** TGF-β2 stimuli induced differentiation of synthetic proliferative human coronary arterial smooth muscle cells with increased expression of contractile marker genes: *TAGLN*, *CALD1* and *CNN1*. *P*-values were obtained from one-way repeated measure ANOVA followed by Dunnett post-hoc test (two-sided). Results are expressed as mean ± SEM. *n* = 9 biological replicates from three independent experiments in each group were used to assess *TAGLN* and *CALDI* gene expression. *n* = 6 biological replicates from three independent experiments were used to measure *CNN1* gene expression. Source data are provided in the source data file. Credit: Part **a** created with http://BioRender.com. Released under a Creative Commons Attribution NonCommercial NoDeriatives 4.0 International License (CC BY NC ND 4.0).

such as thin fibrous caps are associated with high-risk plaques in both the carotid and the coronary arteries[54–56]. Next, one should consider that TGF-ßs also affect other ECM components, besides collagens, such as proteoglycans. TGF-ßs have been shown to induce VSMC proteoglycan synthesis as well as to contribute to proteoglycan side chain elongation, which could likely contribute to lipoprotein retention and plaque formation[57–60]. However, proteoglycan synthesis is beyond the scope of the current study. Finally, compared with previous studies using 10x, a lower number of cells were sequenced, here using Smart Seq 2 technology. Nevertheless, the depth of the Smart Seq 2 technology allowed a distinct clustering of the CD45⁻ cells. As a matter of fact, a post-hoc power analysis confirmed that the number of cells sequenced had enough power (>0.99) to detect the five described CD45⁻ cell clusters.

In summary, the present study shows the important role of fibrous tissue repair in T2D to prevent atherosclerotic complications. We identified a key role of TGF-β2 in fibrous cap formation and that hyperglycemia causes dysregulation of two-way crosstalk between *MMP2*⁺ synthetic and *TGFB2*⁺ contractile SMC within the fibrous cap, reducing TGF-β2 release and VSMC differentiation into contractile *TGFB2*⁺ cells and, in turn, collagen production and fibrous cap thickening. Our findings are highly relevant translationally, as they highlight the importance of glycemic control and the fibrous repair mechanisms as therapeutic targets to prevent atherosclerotic CV complications in T2D.

## Methods
The study followed the Declaration of Helsinki and was approved by the local ethical committee in Lund (472/2005; 2014/904; 2017/89; 2018/63). Written and oral informed consent was given by each patient.

### Study cohort
A total of 219 human carotid plaques (147 plaques from ND patients and 72 from T2D patients; Supplementary Table 8) obtained from the Carotid Plaque Imaging Project biobank (Malmö, Sweden; ClinicalTrials.gov ID NCT05821894) were studied. Three patients, all asymptomatic without diabetes, underwent a second endarterectomy. The indications for surgery were plaques associated with ipsilateral symptoms (transitory ischemic attack, stroke or amaurosis fugax) and >70% stenosis measured by ultrasound, or plaques from patients with no symptoms but stenosis >80%. All patients were preoperatively examined by a neurologist. Cardiovascular risk factors, namely hypertension (systolic blood pressure >140 mm Hg), diabetes and current smoking and use of medications (anti-hypertensive drugs, diabetes treatment and statins) were recorded. Blood samples were collected prior to surgery and fasting lipoproteins (total cholesterol, HDL cholesterol, LDL cholesterol and triglycerides), HbA1c, C-reactive protein were measured. Sex (identified by the Swedish personal number) was included as a clinical variable in the study design. Reported results were, whenever possible, adjusted for sex.

### Sample preparation and histology
For histology and bulk RNA sequencing, plaques were snap-frozen in liquid nitrogen immediately after surgical removal, and two 1 mm

fragments from the most stenotic region were kept for analysis. The remaining tissue was homogenized in 5 mL of a homogenization buffer consisting of 2 mmol/L tris(2-carboxyethyl)phosphine HCl, 1 mmol/L benzamidine, 1 mmol/L Na-orthovanadate, 10 mmol/L Na-glycerophosphate, 50 mmol/L Tris-HCl (pH 7.5), 0.25 mol/L sucrose, 50 mmol/L NaF, 5 mmol/L Na-pyrophosphate, protease inhibitor cocktail (Roche Complete, EDTA-free), and 10 mmol/L phenylmethylsulfonyl fluoride[61]. For the histological analyses, embedded fragments were cryosectioned (8 µm sections), fixed with Histochoice (Amresco, Ohio, USA), dipped in 60% isopropanol and then in 0.4% Oil Red O in (60%) isopropanol (for 20 min) to stain for neutral lipids (Oil red O). Vascular smooth muscle cells (alpha-actin) were stained using a primary antibody monoclonal mouse anti-human smooth muscle actin antibody, clone 1A4 (DakoCytomation, Glostrup, Denmark), diluted 1:50 in 10% rabbit serum, and a secondary antibody biotin rabbit anti-mouse Ig (DakoCytomation, Glostrup, Denmark), diluted 1:200 in 10% of rabbit serum. Hemorrhage (glycophorin A) was assessed using a primary antibody monoclonal mouse anti-human glycophorin A(CD235a) (DakoCytomation, Glostrup, Denmark), diluted 1:400 in 10% rabbit serum, and a secondary antibody biotin rabbit anti-mouse F(ab´)2 (DakoCytomation, Glostrup, Denmark), diluted 1:200 in 10% of rabbit serum and CD68 (macrophages) was measured using a primary monoclonal antibody mouse anti-human CD68, clone KP1 (DakoCytomation, Glostrup, Denmark), diluted 1:100 in 10% rabbit serum, and secondary antibody biotinylated polyclonal Rabbit anti-mouse, Rabbit F(ab')2 (DakoCytomation, Glostrup, Denmark), diluted 1:200 in 10% of rabbit serum. Russell-Movat pentachrome was used to detect plaque collagens. Sections were scanned using a ScanScope Console Version 8.2 (LRI imaging AB, Vista CA, USA) and photographed with Aperio image scope v.8.0 (Aperio, Vista California, USA). The positively stained plaque area was quantified (blindly) using BiopixiQ 2.1.8 (Gothenburg, Sweden). The histological vulnerability index was calculated as the sum of CD68, neutral lipids (Oil red O) and hemorrhage (glycophorin A) stained plaque areas divided by the sum of smooth muscle cell (alpha-actin) and collagen (MOVAT pentachrome) stained areas.

### Growth factors, collagen and matrix metalloproteinase measurements
Sixteen growth factors, all with important roles in cardiovascular disease, were assessed in 218 plaque tissue homogenates, of which 71 patients had T2D. Hepatocyte growth factor, β-nerve growth factor, placental growth factor, vascular endothelial growth factor, platelet-derived growth factor subunit B, stem cell factor, heparin-binding EGF like growth factor, growth differentiation factor, colony stimulating factor-1, fibroblast growth factor, epidermal growth factor and growth hormone were measured in plaque tissue homogenates using a Proximity Extension Assay (PEA; Proseek Multiplex CVD96x96 reagents kit, Olink Bioscience, Uppsala, Sweden) at the Clinical Biomarkers Facility, Science for Life Laboratory, Uppsala. Data were preprocessed for normalization using Olink Wizard for GenEx (Multid

Analyses, Sweden). All data are expressed as arbitrary units. Approximate concentrations of growth factors can be calculated from general calibrator curves available at Olink homepage (http://www.olink.com). Platelet derived growth factor-AB/BB and vascular endothelial growth factor were measured using a human Cytokine/chemokine immunoassay (Millipore Corporation, MA, USA) and analysed with Luminex 100 IS 2.3 (Austin, Texas, USA). Absolute concentrations of TGF-ßs (-β1, -β2 and −β3) were measured using Milliplex Map TGF-β Magnetic Bead 3 Plex Kit (MerckMillipore; cat: TGF-ß MAG-64K-03, Billerica, MA, USA) in non-acidified plaque tissue homogenates to ensure that only free/ active TGF-ß levels were assessed. Plaque levels of growth factors were normalized to plaque wet weight.

Plaque collagen (acid and pepsin soluble collagens types I to V) content was measured in plaque tissue homogenate using Sircol soluble collagen assay (Biocolor, Carrickfergus, Northern Ireland, UK). The assay detects acid and pepsin soluble collagens types I to V. Results were expressed as collagen levels normalized to plaque wet weight. MMP2 and -9 levels were analyzed in plaque tissue homogenate using Mesoscale human MMP ultra-sensitive kit (Mesoscale, Gaithersburg, MD, USA) as per the provider's guide. Results were normalized to plaque wet weight.

## Carotid plaque bulk RNA sequencing

Bulk RNA sequencing data of 22 human carotid plaques from patients with T2D were used to quantify the expression of *TGFB1*, *TGFB2*, *TGFB3* and other VSMC marker genes. Total RNA was extracted from the most stenotic plaque region by standard trizol method and cleaned for Ribosomal RNA using Ribo-Zero™ Magnetic Kit from (Epicentre). Strand-specific RNAseq libraries were prepared using ScriptSeq™ v2 RNA-Seq Library v2 Preparation Kit (Epicentre) and sequenced using high-output kit version 2 on HiSeq2000 and NextSeq platform, Illumina, USA. Reads were aligned to human genome assembly GRCh38 using STAR aligner and quantified by Salmon with gene annotation GENCODE V27[62,63]. Counts were normalized by edgeR, and gene expression was normalized as log2- transformed count per million (CPM)[64]. Subsequently, batch effects of sequencing platforms (Illumina HiSeq2000 and the NextSeq platforms) were adjusted by an empirical Bayes method Combat[65]. The batch-corrected log2CPM was used for further analysis.

## Carotid plaque single-cell RNA sequencing

Carotid plaque cells were isolated using a previously published enzymatic dispersion method with some modifications[66]. Plaque tissue was taken immediately after surgical removal and placed in RPMI 1640 media. To remove circulating cells and red blood cells, plaques were washed with RPMI, and red blood cells were lysed using red blood cell lysis buffer (Red Blood Cell Lysing Buffer Hybri-Max™, Sigma-Aldrich). Subsequently, the plaque tissue was cleaned of calcified areas, minced and placed into an enzymatic digestion cocktail consisting of collagenase type I (400 units/mL, Sigma C9722), elastase type III (5 units/mL, Worthington, LS006365), and DNase (300 units/mL, Sigma D5025), with 1 mg/mL soybean trypsin inhibitor (Sigma T6522), 2.5 µg/mL polymixin B (Sigma), and 2 mM CaCl$_2$, in RPMI medium 1640 with 5% FBS. The suspension was incubated at 37 °C for 30 min with continuous agitation. After incubation, cell suspension was pipetted up and down to break the remaining tissue. Thereafter, the cell suspension was strained using a 100 µm strainer and pelleted by centrifugation at 500×g for 5 min. The cells were then suspended in fresh RPMI 1640 media. For sorting purpose, isolated cells were washed, stained with MitoTracker™ Green FM (ThermoFisher Scientific, Cat# M7514) (50 nM 30 min at RT), FC blocked (Biolegend, Cat# 422302) (1:33 15 min at RT), and subsequently incubated with an antibody cocktail consisting of PE/Cy7 anti-human CD235a (Glycophorin A, Biolegend, Cat# 349111), and APC anti-human CD45 (Biolegend, Cat# 304012) antibodies at 1:100 dilution for 30 min at 4 °C. Thereafter,

cells were washed, resuspended in PBS containing viability dye 7-AAD (Biolegend, Cat# 420403) at 1:100 dilution and processed immediately for FACS sorting using a BD FACSAria III cell sorter (BD Biosciences). CD45$^+$ and CD45$^-$ cells were separately sorted into 384 wells of Smart seq2 plates (Eukaryotic genomic facility, Scilife lab, Stockholm) following user guidelines. The sorted plates were frozen instantly on dry ice and delivered to Scilife lab for sequencing.

Quality check, library preparation and sequencing were performed according to standard methods at Scilife eukaryotic genomic facility according to a published protocol, same as the processing of the raw single-cell RNA sequencing data[67]. Cells were dropped for low-quality libraries less than 10,000 raw reads. Cells were further filtered by the Seurat if the number of detected genes was less than 500 and the percentage of ERCC RNA Spike-In was no more than 15%[68]. Counts were log-normalized, scaled and top 2000 highly variable genes were used for dimensional reduction. As determined by the percentages of variance explained by principal components (ElbowPlot functions of the Seurat[69]), the first seven principal components were used to construct a shared nearest neighbor graph and a non-linear dimensional reduction t-distributed stochastic neighbor embedding (t-SNE) for CD45$^+$ and CD45$^-$ cells, respectively. Cell clusters were then identified by the default Louvain algorithm using "FindClusters" function of the Seurat (resolution = 0.5).

Cell type per cluster was determined by canonical cell markers. Cell types were also predicted using single cell reference data of human vasculature from the Tabula[68] by the "TransferData" function of Seurat[69].

Genes highly expressed in one cluster compared to all other clusters were examined by two-sided Wilcoxon Rank-Sum test using the "FindMarkers" function of Seurat. Genes with an average log-2 transformed fold change above 0.25 and adjusted *p*-value less than 0.05 were considered as differentially expressed genes (DEG). Consequently, an R package ReactomePA was implemented to perform pathway enrichment analysis using these DEGs. A significant enriched pathway had a *p*-value less than 0.05 after Benjamini−Hochberg correction. Module scores for the selected pathway were calculated using "AddModuleScore" function from the Seurat R package (version 3.2.3).

## Validation of the identified cell types in an independent dataset

Publicly accessible single-cell RNA sequencing (scRNA-seq) data from human carotid plaques[70], encompassing both the atherosclerotic core (AC) and the proximal adjacent (PA) region, were utilized to validate our findings. This independent dataset was analyzed through a reproducible pipeline designed for scRNA-seq[71]. Based on gene expression of markers for vascular smooth muscle cells (VSMCs, Supplementary Fig. 7), the cells from clusters 5, 8, 11 and 12 (obtained from the PlaqView pipeline[71]) were suggested as VSMCs in the independent dataset. Focusing on these VSMCs, we examined the expression of main differentially expressed genes (DEGs) of the identified VSMCs clusters (namely contractile, adipocyte-like, synthetic/fibroblast-like and macrophage-like VSMCs) from our dataset (Supplementary Fig. 8). Additionally, we predicted VSMC cell types by using our CD45$^-$ cell types as reference. The prediction score for each cell was obtained by label transfer function of the Seurat (Supplementary Fig. 8).

Gene expression of markers for immune cells such as T-cells (*CD3E*, *CD4*, *CD8A*), NK cells (*NCAM1*, *NKG7*, *XCL1*), myeloid cells (*CD163*, *FCGR1A*, *ITGAX*, *C1QA*, *IL1B*, *CD1C*), B-cells (*CD79A*), plasmacytoid dendritic cells (*CLEC4C*) and mast cells (*KIT*) were examined in this independent data. Based on gene expression of these markers, cells in clusters 0, 1 and 17 (obtained from the PlaqView pipeline) were annotated as T-cells, cluster 6 as NK cells, clusters 3, 4 and 7 as myeloid cells, clusters 10 and 14 as B cells, cluster 18 as plasmacytoid dendritic cells and cluster 16 as mast cells. Gene expression of markers for non-immune cells (VSMC: *ACTA2*, *MYH11*, *TAGLN*; fibroblasts: *PLA2G2A*,

*FBLN1*; endothelial cells: *PECAM1*, *VWF*) were also examined. As previously described, cells from clusters 5, 8, 11 and 12 were annotated as VSMCs, cluster 13 as fibroblast, clusters 2, 9 and 15 as endothelial cells. Next, gene expression of TGFB isoforms was examined in this independent data, split by immune cells (T, NK, Myeloid, pDC, mast) and non-immune cells (VSMC, fibroblast, endothelial cells; Supplementary Fig. 10).

## Validation of the TGF isoform expression in an independent dataset

Using the same dataset as in the cell type validation, the expression of the three *TGFB* isoforms was assessed. When comparing the *TGFB2* expression between the Smart-Seq2 (SS2) single-cell analysis to the 10x single-cell RNA sequencing analysis, we identified that *TGFB2* expression was detected in higher levels by SS2 than by the 10x ($p = 2.2 \times 10^{-32}$). We compared the *TGFB2* expression between the core and proximal regions and found that *TGFB2* was detected to a greater extent in proximal regions (Supplementary Fig. 9a). These were then used to validate the expression of *TGFB2* among the VSMCs. Considering the very low expression of *TGFB* isoforms and heterogeneity of plaques, negative binomial regressions were applied to compare the expression of *TGFB2* between the predicted contractile VSMCs and the rest of the VSMCs within the proximal region for each patient. A fixed-effect meta-analysis employing the inverse-variance method was performed to aggregate the differences in expression between contractile VSMCs and the other VSMCs within proximal regions. Results are shown in Supplementary Fig. 9b.

## Visium spatial transcriptomics of carotid plaques

Plaque sections for Visium spatial transcriptomics were taken from the most stenotic region of the plaques and processed according to the user's guide (CG000239 RevD). OCT embedded plaque sections (10 μm) were placed on Visium tissue sample slides, fixed with methanol, stained with hematoxylin and eosin, and imaged by Scan-Scope Console Version 8.2 (LRI imaging AB, Vista CA, USA). Thereafter, samples were permeabilized for 12 min and libraries were prepared according to the Visium Spatial Gene Expression protocol. Library quality was assayed using a Bioanalyzer High Sensitivity chip (Agilent) and 2.0 pM of library was sequenced on NextSeq 500/550 using high Output Kit v2.5 (150 Cycles) at a sequencing depth of 400 million reads-pair per sample. Sequencing was performed using the following read protocol: read 1: 28 cycles; i7 index read: 10 cycles; i5 index read: 10 cycles; and read 2: 91 cycles.

Raw data for each plaque (FASTQ files) and respective histological images were processed with the Space Ranger software v.1.0.0, which uses STAR v.2.5.1b52 for genome alignment, against the Cell Ranger hg38 reference genome refdata-cellranger-GRCh38-3.0.0, available at: http://cf.10xgenomics.com/supp/cell-exp/refdata-cellranger-GRCh38-3.0.0.tar.gz. Further analyses were done by using the Seurat. Mitochondria DNA encoded genes were removed prior to analysis. After that, low-quality spots were filtered if the number of the detected genes was less than 50. Counts were normalized by a regularized negative binomial model sctransform which accounted for technical artifacts while preserving biological variance. Using an 'anchor'-based integration and all the obtained clusters of CD45$^+$ and CD45$^-$ cells from our single-cell RNAseq analysis as references, prediction scores for each spot for each class of CD45$^+$ and CD45$^-$ cell types were obtained by label transfer of the Seurat, respectively. Cell type corresponding to the top prediction score per spot was assigned to the spot.

## In vitro studies of smooth muscle cell differentiation and wound healing

Human coronary artery smooth muscle cells (HCASMC) were obtained from ThermoFisher Scientific (Cat# C0175C) and cultured in Human Vascular Smooth Muscle Cell Basal Medium supplemented (Cat# M231500, ThermoFisher Scientific) with smooth muscle cell growth supplement (Cat# S00725, ThermoFisher Scientific), 100 U/mL penicillin and 100 μg/mL streptomycin in $CO_2$ incubator at 37 °C. For comparison of genes between two phenotypes, $5 \times 10^5$ proliferating HCASMS were grown to 80% confluency in a 12-well plate and differentiated by transferring into differentiation media (Cat# S0085, ThermoFisher Scientific) as recommended. Following 6 days of differentiation, cells were harvested for mRNA isolation using Qiagen RNeasy kit (Qiagen, Cat# 74106) according to the manufacturer's instructions. For comparisons, mRNA from proliferative HCASMC grown in parallel was isolated.

To study if differentiated HCASMCs induce a phenotypic shift in proliferative HCASMCs two different approaches were used. First, $5 \times 10^5$ proliferating HCASMS were exposed to wasted media from 6 days differentiated contractile HCASMCs for 72 h before cells were harvested and mRNA was isolated. In an alternate and more direct approach, proliferative HCASMC were exposed to 5 ng/mL TGF-ß2 for up to 72 h before cells were harvested for mRNA isolation.

In a separate series of experiments, we studied the effect of hyperglycemic conditions on proliferative HCASMCs migration using a wound healing assay. In brief, $1 \times 10^4$ proliferating HCASMCs were seeded in 24-well plates for 48 h. Cells were serum-deprived for 24 h and a scratch (in the middle of each well) was created in the cell monolayer. Cells were then washed and incubated with fresh basal growth media containing 4.6 or 20 mM glucose. To acquire images from the same field across different time points, a reference marking was made by etching the outer bottom of each plate with a razor blade. The images were acquired at 0, 8, 12 and 16 h post glucose stimulation using a Nikon phase contrast (DS-Fi1) microscope using a 10X objective and NIS-Elements Basic Research software. The percentage of wound closure was calculated on image areas analyzed by MRI-wound healing plug-in in ImageJ software (NIH, USA), using Eq. 1. Data was summarized with images from 8 wells.

$$Wound\ closue\,(\%) = 100\% * \left(1 - A_{T_x}/A_{T_0}\right) \qquad (1)$$

where $A_{T_x}$ and $A_{T_0}$ represented wound areas at respective time points and 0 h post glucose stimulation.

## Quantitative PCR

Total RNA was extracted using the RNeasy Mini Kit (Qiagen, Cat# 74106), according to the manufacturer's instructions. Then, 1 μg of total RNA was reverse-transcribed using a High capacity RNA-to-cDNA kit (ThermoFisher Scientific, Cat# 4387406) as per user guidelines. Gene expressions were examined by quantitative real-time PCR on a QuantStudio 7 Flex instrument (Applied Biosystems/ThermoFisher) using Taqman Fast Advanced master mix (ThermoFisher Scientific, Cat# 4444557) and appropriate Taqman probes (Supplementary Table 9). Relative gene expression was calculated with QuantStudio Software v1.1 (ThermoFisher) using the ΔΔCt method and normalized to GAPDH expression as endogenous control. For comparison of different groups, gene expression levels are expressed as fold change expression compared to control samples.

## Western blotting

Cultured proliferative HCASMCs were lysed directly in 1x Laemmli sample buffer (#1610747, Bio-Rad) and boiled 10 min at 95 °C under intensive shaking to shear DNA. Proteins were separated by SDS-PAGE (using 4–15% Mini-PROTEAN® Criterion TGX Stain-Free gels; #456-8084, Bio-Rad) and transferred to PVDF (Polyvinylidene fluoride) membranes (Bio-Rad). The membrane was blocked with 5% bovine serum albumin in TBST (20 mM Tris, 150 mM NaCl, 0.1% Tween) and incubated overnight at 4 °C with primary Anti-p-SMAD3 (phospho S423+S425; 1:1000, # ab52903, Abcam) or Anti-p-SMAD2 (phospho S467; 1:1000, #ab53100, Abcam). On the following day, the PVDF

membrane was washed with TBST buffer 5 times (5 min each), incubated with anti-rabbit horseradish peroxidase (HRP)-conjugated secondary antibodies (1:1000, #P0448, Dako) and kept at room temperature for 1 h. The membrane was washed with TBST buffer as before. Enhanced chemiluminescent (ECL) substrates (#32106, ThermoFisher) were employed for the detection of HRP enzyme activity.

For re-probing, the antibodies were stripped using a western blot stripping solution (#21059, ThermoFisher) for 10 min. Thereafter, membranes were blocked with 5% bovine serum albumin in TBST and incubated overnight at 4 °C with primary Anti-SMAD3 (1:1000, # ab40854, Abcam) or Anti-SMAD2 (1:1000, # ab40855). On the following day, the PVDF membrane was washed, incubated with anti-rabbit (HRP)-conjugated secondary antibodies for 1 h, washed with TBST as before, and ECL substrates were employed for the detection of HRP enzyme activity. Then the antibodies were stripped again and the PVDF membrane was incubated with Anti-GAPDH (1:5000, #ab8245, Abcam) for loading control for 2 h and incubated with anti-mouse (HRP)-conjugated secondary antibodies (1:1000, #P0447, Dako) for 1 h, washed with TBST as before and ECL substrates were employed for the detection of HRP enzyme activity.

Protein bands were visualized on a ChemiDoc MP instrument (Bio-Rad) and subjected to densitometric quantification using ImageLab v6.1 software (Bio-Rad).

### Plaque cell experiments

Plaque cells were isolated as described above and cultured in RPMI 1640 media supplemented with 10% FBS and penicillin and streptomycin (1%). After 24 h of incubation, media was collected and centrifuged at 500×$g$ for 5 min to remove debris. The clean supernatant was used for the assessment of TGF-β release as described above.

### Zymography of plaque cells to measure MMP activity

To determine MMP2 activity, the supernatant of plaque cells from patients with type 2 diabetes or without diabetes, or the supernatant of HCASMCs stimulated for 48 h with 5 (control), 10, and 20 mM glucose, was first mixed with 2x sample buffer (ThermoFisher Scientific, LC2676) and loaded onto Novex 10% Zymogram Plus (Gelatin) Protein Gels (ThermoFisher Scientific ZY00100BOX) and ran at constant 200 V for 40 min. Afterward, the gel was incubated with 1x Renaturing Buffer (ThermoFisher Scientific, LC2670) for 30 min at room temperature with gentle agitation. The buffer was then replaced with 1x Developing Buffer (ThermoFisher Scientific, LC2671) for 30 min at room temperature, and successively replaced with new 1x Developing Buffer for 24 h at 37 °C. Finally, the gel was rinsed with distilled water and incubated (30 min to 2 h) with Coomassie Brilliant Blue. The areas of protease activity appear as clear bands against a dark background. Bands were visualized on a ChemiDoc MP instrument (Bio-Rad) and subjected to densitometric quantification using ImageJ 1.54f.

### Follow-up

(CV) events (including myocardial infarction, cardiovascular death, amaurosis fugax and stroke (ipsilateral and contralateral events)) for T2D patients were retrieved from the Swedish National Inpatient Health Register (2005–2013) during the follow-up (up to 85 months). Loss of follow-up (due to emigration from Sweden) or death due to other causes than cardiovascular were censored. Kaplan–Meier survival analysis was conducted comparing patients with vulnerability index (VI) in 1st and 2nd tertiles to 3rd tertile, as planned a priori to gain power.

### Multivariate analysis

Orthogonal partial least squares discriminant analysis (OPLS-DA) was performed in SIMCA-P software package (version 17, Umetrics, Umeå, Sweden). Prior to analysis, data was mean-centered and scaled to unit variance (UV) to remove systematic differences in variables caused by differences in their measurement units. Model performance was evaluated by $R^2$ and $Q^2$ values varying from 0 to 1[72]. $R^2$ represents the proportion of variance in T2D explained by the model, whereas $Q^2$ is a measure of the model's predictive ability, obtained through 7-fold cross-validation. Variables responsible for separation between classes were examined from the variable influence on projection (VIP) values of each variable. VIP is defined as the fraction to which each variable explains Y variance and ranks the variables according to their contribution to the model. Hierarchical clustering of variables was performed by using the Euclidean distance matrix and average linkage method by using the R package pheatmap (version 1.0.12).

### Statistics

Mann–Whitney $U$ and Chi-square tests were used for comparisons between continuous and categorical variables and Spearman's rho statistics for assessing correlations between continuous variables. Multiple linear regression was used to examine the components of vulnerability index in plaques from T2D and ND subjects and to identify factors associated with plaque levels of free TGF-ß2. Log rank test was used in Kaplan–Meier survival analysis. Orthogonal partial least squares discriminant analysis (OPLS-DA) was performed in SIMCA-P (version 17, Umetrics, Umeå, Sweden). In this analysis, model performances were evaluated by $R^2$ and $Q^2$ values. ANOVA of the cross-validated residuals (CV-ANOVA) was implemented to examine the significance of the OPLS-DA model. Contribution of variables in classes separations were examined by the variable influence on the projection (variable importance plot; VIP) values of each variable. Hierarchical clustering was implemented using the average linkage method and Euclidean distance in R (version 4.2.2). Statistical analyses were performed on SPSS 22.0 (IBM Corp., Armonk, NY, USA) or GraphPad Prism 9 (GraphPad Software, Boston, USA) unless stated otherwise.

### Reporting summary

Further information on research design is available in the Nature Portfolio Reporting Summary linked to this article.

## Data availability

Individual data regarding living humans cannot be publicly available due to the sensitive nature of the data regulated by European privacy laws (GDPR). It might be accessed for the sole purpose of replicating the procedures and results presented in the article and providing that the data transfer is in agreement with European Union legislation on the general data protection regulation and decisions by the ethical review board of Sweden, the Region Skåne and the Lund University. The datasets are available with restricted access in the Swedish national infrastructure Science for Life Laboratory repository at https://doi.org/10.17044/scilifelab.26063056. This link includes the terms of data accessibility by request to the corresponding author (Andreas.Edsfeldt@med.lu.se) in up to 6 weeks. An independent single-cell dataset from human carotid plaques used in this study is available in the Gene Expression Omnibus database under accession code GSE159677 and can be retrieved by logging in to PlaqView [https://www.plaqview.com/data]. The Cell Ranger hg38 reference genome refdata-cellranger-GRCh38-3.0.0 is available at http://cf.10xgenomics.com/supp/cell-exp/refdata-cellranger-GRCh38-3.0.0.tar.gz. Source data are provided with this paper.

## Code availability

R codes are made available for research use and are archived in Zenodo (https://doi.org/10.5281/zenodo.10420614). Other softwares used in the present study includes SIMCA-P (version 14.1, Umetrics, Sweden), SPSS (version 22.0, IBM Corp., USA), GraphPad Prism (version 8, GraphPad Software, USA), BD FACSDiva (version 9.0.1, BD Biosciences, USA), STAR (version 2.5.1b52), Space Ranger software (version 1.0.0,

10X, USA) and R (version 4.2.2, Austria) packages of Seurat (version 3.2.3), ggforestplot (version 0.1.0), pheatmap (version 1.0.12), edge-R(version 3.40.2), sva (version 3.46.0) and ggplot2 (version 3.4.2).

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

## Acknowledgements

The work was supported by the Swedish Society for Medical Research [A.E., CG-22-0254-H-02], the Swedish Research Council [A.E., 2019-01907; I.G., 2019-01260] Crafoord Foundation [A.E. 20210796], The Swedish Society of Medicine [A.E., SLS-961085], Swedish Heart and Lung Foundation [A.E., 20220044 and 20220284; I.G., 20200403], the Swedish Stroke Association [J.S., S-993166], Hjelt Diabetes Foundation [J.S.], SUS foundations and funds [A.E.; I.G.] and Lund University Diabetes Center (Swedish Research Council–Strategic Research Area Exodiab Dnr 2009-1039 and the Swedish Foundation for Strategic Research Dnr IRC15-0067). The Knut and Alice Wallenberg Foundation, the Medical Faculty at Lund University and Region Skåne are acknowledged for generous financial support [A.E.]. The authors acknowledge support from the National Genomics Infrastructure in Stockholm funded by Science for Life Laboratory, the Knut and Alice Wallenberg Foundation and the Swedish Research Council, and NAISS/Uppsala Multidisciplinary Center for Advanced Computational Science for assistance with massively parallel sequencing and access to the UPPMAX computational infrastructure. The flow cytometry study was performed at the LUDC-Flow Cytometry Core Facility and was supported by the Swedish Research Council, Strategic Research Area Exodiab, Dnr 2009-1039, by the Swedish Foundation for Strategic Research Dnr IRC15-0067, by Region Skåne/(ALF) and by the Infrastructure Grant of Lund University, 2018.

## Author contributions

P.S., M.C., D.A.-S., F.M., M.B., C.T., L.S., M.N., S.R., P.D. and D.E. performed experimental studies and data analyses. E.B. planned a part of the experimental studies. J.S. made statistical analyses. A.P. collected tissues and analyzed data. P.S., J.S., J.N., C.M., I.G. and A.E. drafted the manuscript. I.G. and A.E. supervised, planned and funded the work. P.S., M.C., D.A.-S., F.M., M.B., C.T., L.S., M.N., S.R., P.D., D.E., A.P., E.B., J.N., M.O.-M., C.M., I.G. and A.E. critically supervised the manuscript.

## Funding

## Competing interests

A.E. reports consulting fees from Novo Nordisk, Sanofi and Amgen, but this has not had any relationship with the current study or affected the design/outcome of the study.
