## [Peer Review File · Nature Communications]

REVIEWER COMMENTS

Reviewer #1 (Remarks to the Author):

Nat Comms article on TGF- β , plaques and diabetes

The Transforming Growth Factor (TGF)- β super family of growth factors/cytokines includes most prominently TGF- β 1, 2 and 3. These proteins are very similar in structure having about 400 amino acids each. They are amongst the most active molecules in biology with the agents having many very substantial effects on cells including stimulation and inhibit of cell proliferation, cell differentiation, matrix secretion etc etc.

Cardiovascular disease (heart attacks and strokes) are the largest single cause of premature mortality in the world and the major underlying pathology is atherosclerosis. Atherosclerosis is the slow formation and later rupture of atherosclerotic plaques which generates the life-threatening clinical event.

The role of TGF- β (s) has been highly controversial – because of the large of biological activities and the relevant processes in the formation and rupture of atherosclerotic plaques it has been possible to find evidence supporting a role for TGF- β as being promoting or inhibiting the process of atherosclerosis.

Diabetes (mostly the metabolic disease of Type 2 diabetes) is associated with a substantially higher rate of CV mortality and indeed the majority of people with T2D die of CV disease. The major metabolic abnormality of diabetes is hyperglycaemia however there are multiple mechanisms and pathways through which hyperglycaemia can arise. Perhaps for this reason these are numerous therapeutic interventions which can reduce hyperglycaemia but these are not as efficacious as might be expected. This is due to the different mechanisms causing hyperglycaemia and also probably the role of pleiotropic (anti-atherosclerotic) actions of these various drugs which have been utilised.

The above analysis reflects on the very broad area of atherosclerosis and indicates the need for further research to reveal the elusive mechanisms and also the impact of diabetes and the fact that although diabetes is associated with elevated levels of TGFs the role of TGF- β is certainly not resolved.

The current paper seeks to address the questions outlined above. The studies were conducted in several hundred human samples and in supporting cell culture experiments.

The human samples were carotid artery plaques presumably taken during the process of endarterectomy when the plaques are extracted for carotid arteries in the neck of patients.

A highly suitable suite of experiments has been undertaken and some really novel findings have emerged which certainly clarify areas that have been disputed or controversial for years.

Findings:

This paper provides high quality data which supports these findings:

- Of the TGF isoforms the level of TGF- β 2 correlates with disease status
- TGF beta can regulate the phenotype – contractile versus synthetic – of vascular smooth muscle cells and contractile state VSMCs arise in plaques due to the action of TGF to promote the contractile phenotype
- Synthetic state VSMCs are present in plaques and they secrete TGF- β 2
- Active TGF- β s are released by proteolysis and in plaques the Matrix Metalloproteinase MMP-2 is associated with the release of active TGF- β
- TGF- β s promote the production of collagen (pro-fibrotic) which stabilises plaques
- Hyperglycaemia (diabetes) reduces the release of MMP-2 from VSMCs
- The reduced MMP-2 leads to less TGF- β 2 and less collagen and therefore more thin capped and vulnerable (to rupture) plaques.
- This sequence reflects of the process of atherosclerosis and also on the reasons for the enhanced susceptibility to the deleterious effects of plaque rupture in people with diabetes.
-

Comments

The paper states at several points “ balance between inflammation and tissue repair” – this seems to be a broad ill-defined statement that is far from the level of insight reflected in the paper

as a whole. Does this mean that inflammation is anti- fibrotic and TGF- β is profibrotic? Inflammation is very much more complicated than is suggested by this statement and the use of "inflammation" (meaning the whole of the immune system response) in this paper and this context is very confusing and somewhat misleading and should be removed or clarified.

The major cellular signalling effects of the action of TGF- β are Smad transcription factors so they are vital to understanding the action of TGF- β which is critical to this paper. Smads have 2 signalling areas which can be phosphorylated and activated - the canonical carboxy terminal phosphorylation and the near ubiquitous non-canonical phosphorylation in the central linker region - these are very different responses with different interpretations. The data does not indicate which of the possible epitopes was used - the figures and the text should be adjusted throughout to indicate the actual epitope for the antibody and there should be some short discussion of the role of Smads in TGF- β signalling as it relates to this paper. The source, catalogue number and epitope of the antibody used should be given clearly as it could not be located by this Reviewer. Also in studies of Smads it is preferred that the loading control for Western blots be the unphosphorylated Smad rather than GAPDH (or both is preference).

Furthermore, much of the Smad work in cardiovascular tissue is on Smad2 - the current work is on Smad3 - there should be data given for Smad2 (phosphorylation) and again the implications should be discussed in the paper.

The study has been conducted in carotid artery samples - heart attacks are the major cause of premature mortality in people with diabetes so the implications for studies of carotid arteries for those in coronary arteries should be considered and discussed.

As the paper is on extracellular matrix then one important element is that of proteoglycans and the effect of TGF β - TGF β (acting via Smad phosphorylation) is a very strong activator of the expression of the enzymes and transferases which are rate limiting for the synthesis of glycosaminoglycan chains on proteoglycans (e.g. biglycan) leading to enhanced binding and retention of atherogenic lipoproteins in the vessel wall (and plaques). See the work of Ira Tabas in NY and Thomas Wight in Seattle. A full study of proteoglycans is beyond the scope of this work but the implications for this topic should be addressed in the discussion.

END

Reviewer #3 (Remarks to the Author):

The authors seek to address an important question, i.e. search for the mechanisms of plaque instability in patients with type 2 diabetes (T2D). This work combines the powers of several advanced technologies, including the bulk RNA sequencing, the single cell RNA sequencing (Smart-Seq2 platform), and the spatial transcriptome sequencing (Visium platform from 10x Genomic). The authors proposed that the low levels of TGF β s protein levels in T2D plaques causes the imbalance of TGF β 2+ contractile and MMP2+ synthetic SMC subsets within the fibrous cap region, which may impair fibrous cap formation. However, further experimental evidence is required to support this interesting hypothesis.

Here are the major concerns:

1. This reviewer had a major problem regarding the quality of scRNA-seq data, i.e., the low cell number examined and the potential blood contamination during plaque tissue preparation:
 - (a) One of major aims of current study was to examine the heterogeneity of VSMCs within human plaques by scRNA-seq. However, 855 single cells are too few numbers to address the heterogeneity of VSMCs. It is not crystal clear whether 855 cells include both CD45+ and CD45- cells or it only contains CD45- cells.
 - (b) Although the authors did not report the cell number and cellular composition of CD45+ cells of human plaque scRNA-seq data, their data in Figure 3A shows that monocytes, CD8+ T cells, CD4+ T cells, and NK cells are the major immune cells in their dataset. The low percentages of macrophages and the high percentages of monocytes in human carotid plaques in current study is inconsistent with previous publications by other groups, i.e., Fernandez et al., 2019, Nat Med, <https://doi.org/10.1038/s41591-019-0590-4>; Mokry et al., 2022, Nat Cardiovasc Res.

<https://doi.org/10.1038/s44161-022-00171-0>. This data raises the concern that human carotid plaque tissues used in current study may be contaminated by blood heavily.

2. This reviewer had another major problem regarding the definition of four types of VSMC subsets by scRNA-seq data. The authors claimed that there are VSMCs, fibroblasts, chondrocytes, and endothelial cells in CD45⁻ population in Figure 3A. However, in their Figure 3B, the authors re-clustered all CD45⁻ cells into 5 major subsets, named as 4 VSMC subsets plus one endothelial cell. Why fibroblasts and the chondrocytes disappeared from their t-SNE map in Figure 3B? Are the authors renamed fibroblasts as "synthetic/fibroblast-like VSMCs"? The authors required to show evidence that synthetic/fibroblast-like VSMCs, macrophage-like VSMCs, adipocyte-like VSMCs did not express prototypical markers for fibroblasts, macrophages, and chondrocytes.

3. The authors showed the expression of TGF β 1, TGF β 2, TGF β 3 expression in 4 VSMC subsets in Figure 3D, however, the authors did not show the expression levels of these genes at single cell level, i.e., are contractile VSMCs (cluster 1) express higher levels of TGF β 1, TGF β 2, TGF β 3 mRNA when compared to other VSMC clusters (cluster 2,3,5) in vivo? This is an important missing data. In vitro data shown in Figure 3E may be not relevant to data in Figure 3D.

4. After Visium spatial deep sequencing of human plaques, the authors simplified the complicated cellular landscapes (including all immune and nonimmune subsets) within human plaques into 4 clusters of VSMC. This strategy deserves to be further discussed. To show an unbiased overview of spatial location of all cell subsets within plaques, particularly within the cap region, including VSMCs and other immune and nonimmune cells will be important to atherosclerosis research community. Only by including all cell subsets in the spatial landscape of human plaques, the readers can judge whether the proposed mechanism is still validated.

Small concerns:

1. Please label which tissue is T2D plaque, which is non-T2D plaque in Figure 1A.

2. Please add P value in Figure 6E,

Point by point response to reviewers

for

“Dysregulation of MMP2-dependent TGF-β2 activation in plaque VSMCs impairs fibrous cap formation in diabetes”

Pratibha Singh, Jiangming Sun, Michele Cavallera, Dania Al Sharify, Frank Matthes, Mohammad Barghouth, Christoffer Tengryd, Pontus Dunér, Ana Persson, Lena Sundius, Mihaela Nitulescu, Eva Bengtsson, Sara Rattik, Daniel Engelbertsen, Marju Orho-Melander, Jan Nilsson, Claudia Monaco, Isabel Goncalves, Andreas Edsfeldt

We wish to thank the reviewers and editors for their constructive comments that allowed us to improve the manuscript. We have revised the manuscript according to the reviewers' comments. Please find a detailed response to the comments below. The changes in the manuscript are highlighted in red.

REVIEWER COMMENTS

Reviewer #1 (Remarks to the Author):

Nat Comms article on TGF-β, plaques and diabetes

The Transforming Growth Factor (TGF)-β super family of growth factors/cytokines includes most prominently TGF-β 1, 2 and 3. These proteins are very similar in structure having about 400 amino acids each. They are amongst the most active molecules in biology with the agents having many very substantial effects on cells including stimulation and inhibit of cell proliferation, cell differentiation, matrix secretion etc etc.

Cardiovascular disease (heart attacks and strokes) are the largest single cause of premature mortality in the world and the major underlying pathology is atherosclerosis. Atherosclerosis is the slow formation and later rupture of atherosclerotic plaques which generates the life-threatening clinical event.

The role of TGF-β(s) has been highly controversial – because of the large of biological activities and the relevant processes in the formation and rupture of atherosclerotic plaques it has been possible to find evidence supporting a role for TGF-β as being promoting or inhibiting the process of atherosclerosis.

Diabetes (mostly the metabolic disease of Type 2 diabetes) is associated with a substantially higher rate of CV mortality and indeed the majority of people with T2D die of CV disease. The major metabolic abnormality of diabetes is hyperglycaemia however there are multiple mechanisms and pathways through which hyperglycaemia can arise. Perhaps for this reason these are numerous therapeutic interventions which can reduce hyperglycaemia but these are not as efficacious as might be expected. This is due to the different mechanisms causing hyperglycaemia and also probably the role of pleiotropic (anti-atherosclerotic) actions of these various drugs which have been utilised.

The above analysis reflects on the very broad area of atherosclerosis and indicates the need for further research to reveal the elusive mechanisms and also the impact of diabetes and

the fact that although diabetes is associated with elevated levels of TGFs the role of TGF- β is certainly not resolved.

The current paper seeks to address the questions outlined above. The studies were conducted in several hundred human samples and in supporting cell culture experiments. The human samples were carotid artery plaques presumably taken during the process of endarterectomy when the plaques are extracted for carotid arteries in the neck of patients. A highly suitable suite of experiments has been undertaken and some really novel findings have emerged which certainly clarify areas that have been disputed or controversial for years.

Findings:

This paper provides high quality data which supports these findings:

- Of the TGF isoforms the level of TGF- β 2 correlates with disease status
- TGF beta can regulate the phenotype – contractile versus synthetic – of vascular smooth muscle cells and contractile state VSMCs arise in plaques due to the action of TGF to promote the contractile phenotype
- Synthetic state VSMCs are present in plaques and they secrete TGF- β 2
- Active TGF- β s are released by proteolysis and in plaques the Matrix Metalloproteinase MMP-2 is associated with the release of active TGF- β
- TGF- β s promote the production of collagen (pro-fibrotic) which stabilises plaques
- Hyperglycaemia (diabetes) reduces the release of MMP-2 from VSMCs
- The reduced MMP-2 leads to less TGF- β 2 and less collagen and therefore more thin capped and vulnerable (to rupture) plaques.
- This sequence reflects of the process of atherosclerosis and also on the reasons for the enhanced susceptibility to the deleterious effects of plaque rupture in people with diabetes.

We wish to thank the reviewer for supporting the potential impact of the current findings.

Comments

1. The paper states at several points “ ... balance between inflammation and tissue repair” – this seems to be a broad ill-defined statement that is far from the level of insight reflected in the paper as a whole. Does this mean that inflammation is anti- fibrotic and TGF- β is profibrotic?

The authors agree with the reviewer that this statement could indeed be better defined. As indicated by the reviewer, inflammatory cells and their mediators may also contribute to pro-fibrotic processes. However, in previous literature, the inflammatory process described in atherosclerotic plaques has been most often closely linked to matrix degradation and especially MMP9-driven degradation of the fibrous cap, which has been shown to be a key process underlying acute plaque ruptures¹⁻³. To obtain homeostasis in healthy tissues, inflammation is followed by its resolution leading to the healing or repair, requiring appropriate matrix synthesis. To better reflect the focus of the current study, this statement has been defined in more exact way together with the concept of inflammation.

Please see the introduction, page 3, line 13-18:

“Plaque matrix metalloproteinases, and especially matrix metalloproteinase-9, induced extracellular matrix (ECM) degradation, triggered by pro-inflammatory transcription factors such as NF- κ B or c-Myc, have been extensively studied and suggested as key processes in

plaque ruptures. Yet, recent human plaque studies have not been able to confirm an enhanced inflammatory activity, as measured by the presence of CD68⁺ macrophages or plaque levels of pro-inflammatory cytokines, in human T2D plaques”

2. Inflammation is very much more complicated than is suggested by this statement and the use of “inflammation” (meaning the whole of the immune system response) in this paper and this context is very confusing and somewhat misleading and should be removed or clarified.

We fully agree with the reviewer regarding the complexity of inflammation and with the valid point that our previous use of the word “inflammation” might have been overly simplistic and misleading. Therefore, we have now removed this wording and rather described which specific pro-inflammatory pathways we are referring to.

Please see the introduction page 3 line 13-18:

“Plaque matrix metalloproteinases, and especially matrix metalloproteinase-9, induced extracellular matrix (ECM) degradation, triggered by pro-inflammatory transcription factors such as NF-κB or c-Myc, have been extensively studied and suggested as key processes in plaque ruptures. Yet, recent human plaque studies have not been able to confirm an enhanced inflammatory activity, as measured by the presence of CD68⁺ macrophages or plaque levels of pro-inflammatory cytokines, in human T2D plaques.”

Please see the results section, page 5 line 15-17:

“Next, we explored if an imbalance between markers associated fibrous matrix repair and markers associated with a vulnerable plaque phenotype predicted future CV events among T2D patients using a previously published histological vulnerability index (VI).”

Please see page 15, line 20-24:

“Next, using the previously published histological ratio (vulnerability index) reflecting the balance between vulnerable plaque components (neutral lipids, CD68⁺ cells and haemorrhage) and fibrous repair associated plaque components (collagen and alpha-actin⁺ cells), we showed that T2D individuals with high plaque vulnerability index were at significantly higher risk for future CV events.”

3. The major cellular signalling effects of the action of TGF-β are Smad transcription factors so they are vital to understanding the action of TGF-β which is critical to this paper. Smads have 2 signalling areas which can be phosphorylated and activated - the canonical carboxy terminal phosphorylation and the near ubiquitous non-canonical phosphorylation in the central linker region – these are very different responses with different interpretations. The data does not indicate which of the possible epitopes was used – the figures and the text should be adjusted throughout to indicate the actual epitope for the antibody and there should be some short discussion of the role of Smads in TGF-β signalling as it relates to this paper. The source, catalogue number and epitope of the antibody used should be given clearly as it could not be located by this Reviewer. Also in studies of Smads it is preferred that the loading control for Western blots be the unphosphorylated Smad rather than GAPDH (or both is preference).

Furthermore, much of the Smad work in cardiovascular tissue is on Smad2 – the current

work is on Smad3 – there should be data given for Smad2 (phosphorylation) and again the implications should be discussed in the paper.

As pointed out by the reviewer, the SMAD activation pattern is key to fully understand TGF- β responses, which are dependent on the SMAD proteins phosphorylation sites. In the current study we used an anti-pSMAD3 antibody (Abcam, Cat # AB52903) to detect TGF- β -induced phosphorylation of SMAD3 at two canonical carboxy terminal serine residues, Ser423 and Ser425. The onset of this phosphorylation is expected to trigger SMAD3 dissociation from its receptors to form a complex with SMAD4, which accumulates in the nucleus where it directly binds to DNA and elicit transcriptional responses⁴⁻⁶. As requested by the reviewer, we have now performed additional experiments to normalize the p-SMAD3 to GAPDH and total SMAD3 separately. Importantly, the TGF- β induced phosphorylation of the c-terminal serine 423/425 residues of SMAD3 was not affected by the two different normalization strategies, please see the updated supplementary figure 9:

a)

b)

New Supplementary Fig. 9. TGF- β 1, - β 2 and - β 3 induced Smad3 phosphorylation in synthetic human coronary arterial smooth muscle cells (HCASMC). HCASMC were treated for 15 min with up to 5 ng/ml of TGF- β 1, - β 2, or - β 3. Proteins were extracted by direct lysis in loading buffer, separated by 10 % SDS-PAGE and subjected to Western blotting detecting phospho-Smad3 (Ser 423/425) and GAPDH (left panel) or total SMAD3 (right panel) as loading controls. Densitometric quantification of independent experiments (means \pm SD). Values were normalized to untreated controls. n=3.

Additional experiments were also carried out to explore if the three TGF- β s also contribute to SMAD2 phosphorylation using an anti-pSMAD2 antibody (Abcam, Cat # AB52903) to detect TGF- β -induced phosphorylation of SMAD2 at carboxy terminal serine residue Ser467. Importantly, we confirmed a dose-dependent SMAD2 phosphorylation upon TGF- β stimuli particularly with TGF- β 2 stimuli, as shown in the update figure 4c and 4d.

New figure 4c)

New figure 4d)

New figure 4c and 4d. TGF- β 1, - β 2 and - β 3 induced Smad2 phosphorylation in synthetic human coronary arterial smooth muscle cells (HCASMC). HCASMC were treated for 15 min with up to 5 ng/ml of TGF- β 1, - β 2, or - β 3. Proteins were extracted by direct lysis in loading buffer, separated by 10 % SDS-PAGE and subjected to Western blotting detecting phospho-SMAD2 (Ser467) and GAPDH (left panel) or total SMAD2 (right panel) as loading controls. Densitometric quantification of independent experiments (means \pm SD). Values were normalized to untreated controls. n=3.

The supplementary methods section and results section have now been improved accordingly, please see the supplementary methods section page 9-10 line 15-14, the results section page 9 line 17-21 and the updated figure 4c and 4d as well as supplementary figure 9 (both shown above):

“Cultured HCASMCs were lysed directly in 1x Laemmli sample buffer (#1610747, Bio-Rad) and boiled 10 min at 95°C under intensive shaking to shear DNA. Proteins were separated by SDS-PAGE (using 4–15% Mini-PROTEAN® Criterion TGX Stain-Free gels; #456-8084, Bio-Rad) and transferred to PVDF membranes (Bio-Rad). The membrane was blocked with 5 % bovine serum albumin in TBST (20 mM Tris, 150 mM NaCl, 0.1% Tween) and incubated overnight at 4 °C with primary Anti-p-SMAD3 (phospho S423 + S425) (1:1000, # ab52903, Abcam) or Anti-p-SMAD2 (phospho S467) (1:1000, #ab53100, Abcam). On the following day, the PVDF membrane was washed with TBST buffer 5 times (5 min each), incubated with anti-rabbit horseradish peroxidase (HRP)-conjugated secondary antibodies (1:1000, #P0448, Dako) and kept at room temperature for 1 h. The membrane was washed with TBST buffer as before. Enhanced

chemiluminescent (ECL) substrates (#32106, ThermoFisher) were employed for the detection of HRP enzyme activity.

For re-probing, the antibodies were stripped using WB stripping solution (# 21059, ThermoFisher) for 10 minutes then membrane was blocked with 5 % bovine serum albumin in TBST and incubated overnight at 4 °C with primary Anti-SMAD3 (1:1000, # ab40854, Abcam) or Anti-SMAD2 (1:1000, # ab40855). On the following day, the PVDF membrane was washed, incubated with anti-rabbit (HRP)-conjugated secondary antibodies for 1 h, washed with TBST as before and ECL substrates were employed for the detection of HRP enzyme activity. Then the antibodies were stripped again and the PVDF membrane was incubated with Anti-GAPDH (1:5000, #ab8245, Abcam) for loading control for 2 h and incubated with anti-mouse (HRP)-conjugated secondary antibodies (1:1000, #P0447, Dako) for 1 h, washed with TBST as before and ECL substrates were employed for the detection of HRP enzyme activity.

Protein bands were visualized on a ChemiDoc MP instrument (Bio-Rad) and subjected to densitometric quantification using ImageLab v6.1 software (Bio-Rad)."

"In order to assess the effect of TGF- β 2 on collagen formation, synthetic/proliferative HCASMC were used. First, we concluded that stimulation of synthetic/proliferative HCASMC with increasing concentration of TGF- β s resulted in increased phosphorylation of SMAD2 (Ser467) and SMAD3 (Ser423/425; Fig. 4c-d and Supplementary Fig. 9)."

4. The study has been conducted in carotid artery samples – heart attacks are the major cause of premature mortality in people with diabetes so the implications for studies of carotid arteries for those in coronary arteries should be considered and discussed.

We thank the reviewer for pointing this out and we fully agree that this needs to be better clarified. As mentioned by the reviewer, myocardial infarction is the most common cause of premature death among individuals with T2D. However, the risk of suffering from ischemic stroke is also clearly increased among individuals with diabetes compared to patients without diabetes suffering from other cardiovascular risk factors^{7,8}. Together, these data show that both atherosclerosis related cardio- and cerebrovascular complications occur much more frequently among individuals with T2D, providing evidence for a common link in the formation of high risk atherosclerotic plaques.

Even though there are morphological differences comparing carotid and coronary plaques, key features such as thin fibrous caps or reduced collagen content are associated with high-risk plaques in both vascular territories⁹⁻¹¹.

Furthermore, the aim of the present study was to assess fibrous tissue repair processes in a translational approach, combining tissue analyses (including morphological tissue structure analyses) with *in vitro* experiments and also clinical follow-up analyses. This approach is currently only possible to perform using carotid plaques as these plaques are removed from living subjects (which is needed for the follow-up analyses) without disrupting the plaque morphology. Most coronary plaques are not removed by surgery nowadays, but rather stented or by passed, staying in the patients. Additionally, the coronary lesions are much smaller in size, not providing enough material to allow the advanced and demanding techniques we performed.

The introduction and the discussion have been improved accordingly, please see the introduction page 3 line 2-10 and the discussion page 20-21 line 22-2:

“Individuals with type 2 diabetes (T2D) are at high risk to suffer from both cardio- and cerebrovascular complications (as myocardial infarctions and ischemic strokes) and premature death due to an early aggressive atherosclerotic disease. Considering the increasing prevalence of T2D, the prevention of cardiovascular complications among individuals with T2D has become one of today’s greatest challenges in medicine.

The aggravated atherosclerotic plaque formation in T2D, leading to a greater plaque burden and more frequent plaque ruptures, is well described and despite our clinical advances in treating cardiovascular disease, patients with T2D and cardiovascular risk factors still have a clearly increased risk to suffer from cerebro- or cardiovascular events.”

“There are limitations that need to be considered. First, the current study only examined atherosclerotic plaque biology in plaques from the carotid arteries. Therefore, these findings may not fully reflect atherosclerotic plaque biology in other vascular territories. However, even if plaque morphology may differ between plaques in the coronary and the carotid arteries, key features such as thin fibrous caps are associated with high-risk plaques in both the carotid and the coronary arteries.”

5. As the paper is on extracellular matrix then one important element is that of proteoglycans and the effect of TGF β – TGF β (acting via Smad phosphorylation) is a very strong activator of the expression of the enzymes and transferases which are rate limiting for the synthesis of glycosaminoglycan chains on proteoglycans (e.g. biglycan) leading to enhanced binding and retention of atherogenic lipoproteins in the vessel wall (and plaques). See the work of Ira Tabas in NY and Thomas Wight in Seattle. A full study of proteoglycans is beyond the scope of this work but the implications for this topic should be addressed in the discussion.

We appreciated the question raised by the reviewer. We agree that covering a full story of proteoglycans is beyond the scope of the current study and would need to be addressed separately. TGF- β s regulate proteoglycan synthesis either by inducing biglycan expression or by stimulating hyperelongation of glycosaminoglycans chains with increased affinity for LDL, potentially causing increased retention of lipoproteins in the arterial wall¹²⁻¹⁵. In line with this, TGF- β neutralizing antibody suppressing biglycan expression is shown to reduce plaque formation in mice¹⁶. The discussion has now been improved in accordance with the reviewers’ suggestion, please see the discussion page 21 line 2-7:

“Next, one should consider that TGF- β s also affect other ECM components, besides collagens, such as proteoglycans. TGF- β s have been shown to induce VSMC proteoglycan synthesis as well as to contribute to proteoglycan side chain elongation, which could likely contribute to lipoprotein retention and plaque formation. However, proteoglycan synthesis is beyond the scope of the current study and would need to be addressed in future studies.”

Reviewer #3 (Remarks to the Author):

The authors seek to address an important question, i.e. search for the mechanisms of plaque instability in patients with type 2 diabetes (T2D). This work combines the powers of several advanced technologies, including the bulk RNA sequencing, the single cell RNA sequencing (Smart-Seq2 platform), and the spatial transcriptome sequencing (Visium platform from 10x Genomic). The authors proposed that the low levels of TGFβs protein levels in T2D plaques causes the imbalance of TGFβ2+ contractile and MMP2+ synthetic SMC subsets within the fibrous cap region, which may impair fibrous cap formation. However, further experimental evidence is required to support this interesting hypothesis.

Here are the major concerns:

1. This reviewer had a major problem regarding the quality of scRNA-seq data, i.e., the low cell number examined and the potential blood contamination during plaque tissue preparation:

(a) One of major aims of current study was to examine the heterogeneity of VSMCs within human plaques by scRNA-seq. However, 855 single cells are too few numbers to address the heterogeneity of VSMCs. It is not crystal clear whether 855 cells include both CD45+ and CD45- cells or it only contains CD45- cells.

As pointed out by the reviewer the present study included less cells than previous studies focusing on CD45⁺ plaque cells using the 10x sequencing technique. However, the aim of the current study was to perform deep sequencing of human atherosclerotic plaque cells to uncover in-depth characteristics of the less characterized CD45⁻ plaque cells. Therefore, SmartSeq-2 (SS2) technology was used, instead of the commonly used 10x technology. SS2 is superior to 10x in detection of genes even with low expression levels, splicing junctions, and cell-cell interactions, which are elemental for either cellular phenotype characterization or for uncovering new subpopulation of cells¹⁷.

For a fair comparison of this claim, we compared the sensitivity (median number of discovered genes per cell; Transcripts per million (TPM) > 0 or Unique molecular identifier (UMI) > 0) of our CD45⁻ cell dataset generated using SS2 with the data set of CD45⁻ cells from a previous study by Pan. et al 2020 which was generated using 10x on human carotid plaque cells¹⁸.

As shown in figure 1 for reviewers, the SS2 technology allowed us to capture a significantly higher number of genes per CD45⁻ cell compared to 10X technology.

Figure 1 for reviewers. Higher number genes are detected in CD45⁻ human atherosclerotic plaque cells using the SmartSeq2 sequencing (SS2 (ours)) technique compared to 10x¹⁸. Horizontal lines indicate the median values of the detected genes.

Therefore, even with a lower number of sequenced cells, this plate-based SS2 approach gained a better resolution and gene coverage which helped us to uncover and characterise 5 very distinct CD45⁻ cell clusters (featureplot demonstrating cluster specific differentially expressed genes (DEGs) in figure 2 for reviewers).

It cannot be excluded that adding even more cells could provide an even higher resolution of VSMC phenotypes. However, in order to confirm if the number of sequenced cells provided enough power to accurately identify these 5 clusters of CD45⁻ cells, an additional post-hoc power analysis was performed. Importantly, as seen in figure 3 for the reviewers, the number of sequenced CD45⁻ cells did provide enough power (>0.99) to detect 5 cell clusters if the minimum cells per cluster was greater than 20, and the minimum cell type frequency was greater than 0.1 (figure 3 for reviewers; <https://satijalab.org/howmanycells/>). In agreement, a minimum number cells of 56 and a minimum frequency of 0.11 per cluster were observed in our CD45⁻ cell clusters (n=489).

Figure 3 for reviewers. Power analysis shows the required number of cells, assuming the presence of 5 cell clusters, with a minimum desired count of 20 cells per cluster and a minimum frequency of 0.1.

Finally, using the Plaqview clustering analysis on a publicly available ScRNA sequencing data of human carotid plaques^{19,20}, 4 VSMC clusters were observed. Furthermore, the expression of cluster specific DEGs from our VSMC clusters (MYH11-contractile VSMC; ABCC9-adipocyte-like VSMC; LUM- synthetic/fibroblast-like VSMC; CCND1-macrophage-like VSMC; figure 4 for reviewers) could be replicated in this independent dataset.

Figure 4 for reviewers. Expressions of the four identified VSMC clusters specific differentially expressed genes (DEGs). a) Clusters obtained from clustering analysis by the Plaqview on a single cell dataset of human carotid plaque cells (51 981 cells). Clusters 5, 8, 11 and 12 are suggested as vascular smooth muscle cells based on cell marker expressions and cell type predictions. b) Gene expression of cluster specific DEGs (MYH11, ABCC9, LUM, CCND1). Data was retrieved from the Plaqview (<http://plaqviewv2.uvad.cos.io/>)

In summary, these additional pieces of evidence substantiate the identification of the five CD45⁻ cell cluster described by our single-cell analysis of human carotid plaques. These findings are also further supported by previous murine and *in vitro* studies²¹.

This information has now been added to the manuscript, please see the discussion page 21, line 7-9:

“Finally, comparing with previous studies using 10x, a lower number of cells were sequenced, here using Smart Seq 2 technology. Nevertheless, the depth of the Smart Seq 2 technology allowed a distinct clustering of the CD45⁻ cells. As a matter of fact, a post-hoc power analysis confirmed that the number of cells sequenced had enough power (>0.99) to detect the five described CD45⁻ cell clusters.”

The total number of CD45⁺ (n=366) and CD45⁻ cells (n=489) and this has been made clear in the manuscript, please see page 6-7 line 25-2:

“To this end, single cell deep RNA sequencing (ScRNAseq) data of live human carotid plaque CD45⁺ (n=366) and CD45⁻ cells (n=489) was performed (Fig. 3; gating strategy is provided in Supplementary Figure 2).”

(b) Although the authors did not report the cell number and cellular composition of CD45+ cells of human plaque scRNA-seq data, their data in Figure 3A shows that monocytes, CD8+ T cells, CD4+ T cells, and NK cells are the major immune cells in their dataset. The low percentages of macrophages and the high percentages of monocytes in human carotid plaques in current study is inconsistent with previous publications by other groups, i.e., Fernandez et al., 2019, Nat Med, <https://doi.org/10.1038/s41591-019-0590-4>; Mokry et al., 2022, Nat Cardiovasc Res. <https://doi.org/10.1038/s44161-022-00171-0>. This data raises the concern that human carotid plaque tissues used in current study may be contaminated by blood heavily.

We thank the reviewer for raising this important question. We agree that it could indeed be a potential issue that the analysis could be influenced by cells from the circulating blood. However, in order to avoid this issue, several steps to remove cells from the circulating blood were included in the cell isolation and sorting protocols. First, plaques were flushed with saline upon surgical removal. Second, before cell isolation started the plaque was washed in Roswell Park Memorial Institute (RPMI) 1640 and potential erythrocytes bound to the endothelial surface were lysed using red blood cell lysis buffer and then the plaque was again washed in RPMI 1640. Third, an erythrocyte marker (CD235) was added to the antibodies used to stain the plaque cell in order to gate out potential red blood cells left in the cell isolates from the CD45⁺ cells (supplementary figure 2).

Finally, we confirmed that the erythrocyte markers glycophorin A/B encoded by genes GYPA and GYPB were not detectable in our sequenced single cell dataset verifying our claim of no presence of blood cell contaminants in the sequenced cells (figure 5 for reviewers).

Figure 5 for reviewers. A) tSNE plots visualizing that none of the CD45⁺ human carotid plaque cells were recognized as erythrocytes using reference cell types from Tabula Vasculature. B) Gene expression of glycophorin A (GYPA) and glycophorin B (GYPB).

As also pointed out by the reviewer, the majority of the CD45⁺ myeloid cells in our previous Figure 3A were predicted to be monocytes. However, the cell types presented in the previous Figure 3A were based only on prediction, which in turn is dependent on the cell types presented in the reference data and their similarities with the query data. Therefore, the ideal prediction would be based on reference data from the same tissue as the query data.

However, due to the limited number of publicly available comprehensive human atherosclerotic plaque cell data sets (including both CD45⁺ and CD45⁻ cells), the reference data in our previous analysis consisted of 259 RNA-seq samples of pure stroma and immune cells (Blueprint and ENCODE)²²⁻²⁴. This could likely explain why cells were not properly annotated to the actual cell types. As shown in a previous study by Depuydt *et al.*, there is a major overlap between macrophage and monocyte markers in the human plaque²⁵. The dataset used for prediction is, therefore, likely to have a major influence on prediction of these cells as macrophages or monocytes. If we use the Tabula Vasculature (used for cell type annotation in a collection of cardiovascular single-cell genomic datasets, www.plaqview.com) instead of Blueprint and ENCODE as a reference, all monocytes are recognized as macrophages²⁶. Furthermore, to confirm this prediction of macrophages and dendritic cells using Tabula as reference, we compared the overlap of macrophage and dendritic cell markers using recent human CD45⁺ plaque cell studies as reference^{27,28}. In support for our improved analysis of macrophages and dendritic cells, we confirmed a major overlap of macrophage and dendritic cell genes in our predicted clusters of macrophages and dendritic cells (figure 6 for the reviewers).

Figure6 for reviewers. t-SNE plots showing that the expression of macrophage and dendritic cell markers in human CD45⁺ cells confirm that the majority of the myeloid cells are macrophages.

Considering this potential limitation of cell type prediction based on the reference data sets, we now performed clustering analysis on the CD45⁺ and CD45⁻ cells, separately. Cell type per cluster was then annotated by gene expression of cell makers, in addition to the cell type prediction (using Tabula Vasculature as reference). Please see the updated figure 3a:

New figure 3a. Single cell RNA sequencing analysis of human atherosclerotic carotid plaque showing the different cell types identified among CD45⁺ and CD45⁻ and their expression of the three TGFβ isoforms. NK, natural killer; DC, dendritic cell; VSMC, vascular smooth muscle cell.

As shown in Figure 3A in the revised manuscript, we identified 5 main clusters of CD45⁺ and 5 main clusters of CD45⁻ cells. Out of the 366 CD45⁺ positive cells included in our SS2 sequencing, 193 cells were identified as T-cells, 94 cells as macrophages, 46 cells as NK cells and 33 cells as dendritic cells. The number of macrophages accounted for about 25% of all CD45⁺ cells. According to Fernandez *et al.*, macrophages account for approximately 13% (3-28%) of all CD45⁺ cells whereas Depuyt *et al.* identified that the myeloid cells account for ~24% (after excluding one cluster of apoptotic myeloid and T cells)^{25,29}. In further support, we previously showed, using mass cytometry, that macrophages represent 22% of all CD45⁺ plaque cells²⁷. Altogether, these previous studies suggest that macrophages accounts for 16%–25% of all CD45⁺ live plaque cells, which is in line with our current study.

In summary, the sequenced CD45⁺ cells included in the current study are likely not affected by blood cell contamination and according to the updated and improved analyses the majority of myeloid cell are recognised as macrophages, which is in line with previous studies focusing on human carotid plaque CD45⁺ cells.

This information has now been added to the supplementary methods section and the results section, please see supplementary methods page 4-5 line 25-8 and page 5-6 line 24-11, the results section page 6-7 line 25-3 and page 7 line 15-19, the updated figure 3a (shown above) and the new supplementary figure 4 (shown below):

“In brief, carotid plaque was taken immediately after surgical removal and placed in RPMI 1640 media. To remove circulating cells and red blood cells, plaques were washed with RPMI and red blood cells were lysed using red blood cell lysis buffer (Red Blood Cell Lysing Buffer Hybri-Max™, Sigma-Aldrich). Subsequently, plaque tissue was cleaned of calcified areas, minced and placed into an enzymatic digestion cocktail consisting of collagenase type I (400 units/ml, Sigma C9722), elastase type III (5 units/ml, Worthington, LS006365), and DNase (300 units/ml, Sigma D5025), with 1 mg/ml soybean trypsin inhibitor (Sigma T6522), 2.5 μg/ml polymixin B (Sigma), and 2 mM CaCl₂, in RPMI medium 1640 with 5% FBS. “

“Cells were dropped for low-quality libraries less than 10,000 raw reads. Cells were further filtered by the Seurat if number of the detected genes <500 and percentage of ERCC RNA Spike-

In >15%. Counts were log-normalized, scaled and top 2000 highly variable genes were used for dimensional reduction. As determined by the percentages of variance explained by principal components (ElbowPlot functions of the Seurat), the first seven principal components were used to construct a shared nearest neighbour graph and a non-linear dimensional reduction t-distributed stochastic neighbour embedding (t-SNE) for CD45⁺ and CD45⁻ cells, respectively. Cell clusters were then identified by the default Louvain algorithm using "FindClusters" function of the Seurat (resolution=0.5). Cell type per cluster was determined by canonical cell markers. Cell types were also predicted using single cell reference data of human vasculature from the Tabula by the TransferData function of the Seurat (resolution=0.5)."

"To this end, single cell deep RNA sequencing (ScRNAseq) data of live human carotid plaque CD45⁺ (n=366) and CD45⁻ cells (n=489) was performed (Fig. 3; gating strategy is provided in Supplementary Fig. 2). CD45⁻ cells were the main source of TGFB2 and -B3 whereas TGFB1 was expressed evenly by a vast majority of plaque cells (Fig. 3a)."

"Of all CD45⁻ cells from the ScRNAseq cell analysis, 24.5% were identified as contractile VSMC, 23.1% as adipocyte-like VSMC, 22.1% as synthetic/fibroblast-like VSMC and 11.5% as macrophage-like VSMC (Fig. 3a and 3b). Of all CD45⁺ cells, 52.7% were T-cells, 12.6% were natural killer cells, 9.0% were dendritic cells and 25.8% were macrophages (macrophage 1, 18.6% and macrophage 2, 7.1%, respectively; Fig. 3a and Supplementary Fig. 4a and 4b)."

New supplementary figure 4. Clustering analysis of CD45⁺ carotid plaque cells. A) CD45⁺ Cell clusters and B) expressions of cell marker genes. Cluster 0, T-cells; Cluster 1, macrophage 1; Cluster 2, natural killer cells; Cluster 3, dendritic cells; Cluster 4, Macrophage 2.

2. This reviewer had another major problem regarding the definition of four types of VSMC subsets by scRNA-seq data. The authors claimed that there are VSMCs, fibroblasts, chondrocytes, and endothelial cells in CD45⁻ population in Figure 3A. However, in their Figure 3B, the authors re-clustered all CD45⁻ cells into 5 major subsets, named as 4 VSMC subsets plus one endothelial cell. Why fibroblasts and the chondrocytes disappeared from their t-SNE map in Figure 3B? Are the authors renamed fibroblasts as “synthetic/fibroblast-like VSMCs”? The authors required to show evidence that synthetic/fibroblast-like VSMCs, macrophage-like VSMCs, adipocyte-like VSMCs did not express **prototypical markers** for fibroblasts, macrophages, and chondrocytes.

We agree with the reviewer that this could have been clearer in the previous version of the manuscript. As mentioned in the response to question 2 the described cell types shown in Figure 3A were identified only by prediction using reference dataset Blueprint and ENCODE, consisting of 259 RNA-seq samples of pure stroma and immune cells²²⁻²⁴. As the prediction of particular cell types are dependent on the cell types identified in the reference data there will be differences when comparing predicted cell types to the more granular clustering analysis. To avoid misinterpretation of Figure 3a and Figure 3b we have now updated and improved our analysis. In the current version of the manuscript, we instead performed clustering analyses on CD45⁺ and CD45⁻ cells separately. Cell types per cluster were then annotated by gene expression of cell markers, in addition to the cell type prediction using Tabula Vasculature as reference²⁶. Please see the updated Figure 3a below.

New figure 3A. Single cell RNA sequencing analysis of human atherosclerotic carotid plaque showing the different cell types identified among CD45⁺ and CD45⁻ and their expression of the three TGFB isoforms. NK, natural killer; DC, dendritic cell; VSMC, vascular smooth muscle cell.

Moreover, we did not observe gene expression of fibroblast specific marker DPEP1 in the synthetic/fibroblast-like VSMC cluster. Neither did we identify any convincing overlapping cluster specific expression of the chondrocyte markers EPYC, LEP or COL10A1, the fibroblast markers THY1, DPEP1 and PDGFRA or the macrophage markers ITGAX, ITGAM, CD163, PTPRC, ARG1 or NOS2 (please see the new suppl. figure 5).

Supplementary figure 5. T-SNE plots showing potential overlap of gene expression of prototypical markers for macrophages, adipocytes and fibroblasts in the 5 identified CD45⁻ cell clusters.

The manuscript has been improved accordingly, please the results section page 7 line 20-22, the new Figure 3a and supplementary figure 5 (both shown above):

“To further validate that the four VSMC clusters were not fibroblasts, macrophages or chondrocytes, we investigated the overlap of genes commonly used as prototypical cluster markers for these cell types, but no major overlaps were found (Supplementary Fig. 5).”

3. The authors showed the expression of TGFβ1, TGFβ2, TGFβ3 expression in 4 VSMC subsets in Figure 3D, however, the authors did not show the expression levels of these genes at single cell level, i.e., are contractile VSMCs (cluster 1) express higher levels of TGFβ1, TGFβ2, TGFβ3 mRNA when compared to other VSMC clusters (cluster 2,3,5) in vivo? This is an important missing data. In vitro data shown in Figure 3E may be not relevant to data in Figure 3D.

We are grateful for this constructive comment from the reviewer. The suggested analysis has been performed and is shown in Figure 3d, the *TGFB2*⁺ cells were more frequent among contractile VSMC than the rest of the VSMCs (adipocyte-like, synthetic/fibroblast-like VSMC, macrophage-like). *TGFB1* and *TGFB2* expressions were significantly higher in contractile VSMC compared to the three other clusters of VSMCs, whereas no significant differences in *TGFB3* expression was identified comparing the four VSMC clusters (Supplementary Fig. 6).

This information has now been added to the manuscript, please see the results section page 8 line 2-5 and the supplementary figure 6.

“Also when comparing the *TGFB2* expression among *TGFB*⁺ cells, we identified that *TGFB2* expression was significantly higher in contractile VSMC compared to the other three VSMC clusters ($p=0.002$; Supplementary Fig. 6).”

Supplementary figure 6. Comparing gene expression of TGF isoforms in the contractile VSMC (cluster 1) and other VSMC clusters (clusters 2, 3 and 5). *TGFB1* and *TGFB2* expressions were significantly higher in contractile VSMC compared to the other three VSMC clusters. No significant difference between expression of *TGFB3* was observed. VSMC: Vascular smooth muscle cells.

4. After Visium spatial deep sequencing of human plaques, the authors simplified the complicated cellular landscapes (including all immune and nonimmune subsets) within human plaques into 4 clusters of VSMC. This strategy deserves to be further discussed. To show an unbiased overview of spatial location of all cell subsets within plaques, particularly within the cap region, including VSMCs and other immune and nonimmune cells will be important to atherosclerosis research community. Only by including all cell subsets in the spatial landscape of human plaques, the readers can judge whether the proposed mechanism is still validated.

In the manuscript we focus on the fibrous repair process and the role of TGF-beta. Based on the single cell analysis and our in vitro studies this process is mainly dependent on the CD45⁻ cells. Considering the focus of the study, we focused on clustering only the CD45⁻ spots to explore potential interactions between CD45⁻ cells.

Furthermore, even though the method provides a high spatial resolution of the atherosclerotic plaque, this technique does not achieve single cell resolution and may thereby introduce challenges for cell identification³⁰. The analysis was thus intentionally separated in CD45⁺ and CD45⁻ spots (PTPRC), in accordance with the ScRNA sequencing analysis approach.

Nevertheless, as suggested by the reviewer, we have now mapped both CD45⁺ and CD45⁻ cell phenotypes without considering if the spots were PTPRC⁺ or PTPRC⁻.

Even when considering both PTPRC⁺ and PTPRC⁻ spots, our cell cluster prediction confirms the presence of the four clusters of VSMCs and demonstrates the close proximity of the contractile and synthetic VSMC phenotypes within the fibrous cap.

Please find the new version of mapped cell types using both CD45⁺ and CD45⁻ cells and PTPRC⁺ and PTPRC⁻ cells below. This has now been added as supplementary figure 13 and the text has been updated accordingly.

The manuscript has been updated accordingly, please page 12 line 14-15:

“Macrophages were located close to the core whereas T-cells were broadly distributed (Supplementary Fig. 13).”

Supplementary figure 13. The top predicted cell type per spot (based on the single cell RNA sequencing) showing the morphological distribution of both $PTPRC^+$ and $PTPRC^-$ cell phenotypes in a human carotid plaque. VSMC: vascular smooth muscle cells; EC: Endothelial cells; NK: Natural killer cells; DC: Dendritic cells, H&E: Haematoxylin and eosin. Scale bar 500um. Blue dotted line marks the lumen. Red dotted line marks the fibrous cap.

Small concerns:

1. Please label which tissue is T2D plaque, which is non-T2D plaque in Figure 1A.

The figure has been improved accordingly, please see the updated figure 1A.

2. Please add P value in Figure 6E,

The p-values have now been added, please see the updated figure 6E.

References:

1. Li, T., *et al.* The Role of Matrix Metalloproteinase-9 in Atherosclerotic Plaque Instability. *Mediators Inflamm* **2020**, 3872367 (2020).
2. Monaco, C., *et al.* Toll-like receptor-2 mediates inflammation and matrix degradation in human atherosclerosis. *Circulation* **120**, 2462-2469 (2009).
3. Sun, J., *et al.* Spatial Transcriptional Mapping Reveals Site-Specific Pathways Underlying Human Atherosclerotic Plaque Rupture. *J Am Coll Cardiol* **81**, 2213-2227 (2023).
4. Feng, X.H. & Derynck, R. Specificity and versatility in tgf-beta signaling through Smads. *Annu Rev Cell Dev Biol* **21**, 659-693 (2005).
5. Schmierer, B. & Hill, C.S. TGFbeta-SMAD signal transduction: molecular specificity and functional flexibility. *Nat Rev Mol Cell Biol* **8**, 970-982 (2007).
6. Schmierer, B. & Hill, C.S. Kinetic analysis of Smad nucleocytoplasmic shuttling reveals a mechanism for transforming growth factor beta-dependent nuclear accumulation of Smads. *Mol Cell Biol* **25**, 9845-9858 (2005).
7. Emerging Risk Factors, C., *et al.* Diabetes mellitus, fasting blood glucose concentration, and risk of vascular disease: a collaborative meta-analysis of 102 prospective studies. *Lancet* **375**, 2215-2222 (2010).
8. Rawshani, A., *et al.* Risk Factors, Mortality, and Cardiovascular Outcomes in Patients with Type 2 Diabetes. *N Engl J Med* **379**, 633-644 (2018).
9. Gupta, A., *et al.* Carotid plaque MRI and stroke risk: a systematic review and meta-analysis. *Stroke* **44**, 3071-3077 (2013).
10. Marnane, M., *et al.* Plaque inflammation and unstable morphology are associated with early stroke recurrence in symptomatic carotid stenosis. *Stroke* **45**, 801-806 (2014).
11. Tian, J., *et al.* Distinct morphological features of ruptured culprit plaque for acute coronary events compared to those with silent rupture and thin-cap fibroatheroma: a combined optical coherence tomography and intravascular ultrasound study. *J Am Coll Cardiol* **63**, 2209-2216 (2014).
12. Little, P.J., Tannock, L., Olin, K.L., Chait, A. & Wight, T.N. Proteoglycans synthesized by arterial smooth muscle cells in the presence of transforming growth factor-beta1 exhibit increased binding to LDLs. *Arterioscler Thromb Vasc Biol* **22**, 55-60 (2002).
13. Schonherr, E., Jarvelainen, H.T., Sandell, L.J. & Wight, T.N. Effects of platelet-derived growth factor and transforming growth factor-beta 1 on the synthesis of a large versican-like chondroitin sulfate proteoglycan by arterial smooth muscle cells. *J Biol Chem* **266**, 17640-17647 (1991).
14. Dadlani, H., Ballinger, M.L., Osman, N., Getachew, R. & Little, P.J. Smad and p38 MAP kinase-mediated signaling of proteoglycan synthesis in vascular smooth muscle. *J Biol Chem* **283**, 7844-7852 (2008).
15. Burch, M.L., *et al.* TGF-beta stimulates biglycan synthesis via p38 and ERK phosphorylation of the linker region of Smad2. *Cell Mol Life Sci* **67**, 2077-2090 (2010).
16. Tang, T., *et al.* Prevention of TGFbeta induction attenuates angII-stimulated vascular biglycan and atherosclerosis in Ldlr-/- mice. *J Lipid Res* **54**, 2255-2264 (2013).
17. Wang, X., He, Y., Zhang, Q., Ren, X. & Zhang, Z. Direct Comparative Analyses of 10X Genomics Chromium and Smart-seq2. *Genomics Proteomics Bioinformatics* **19**, 253-266 (2021).

18. Pan, H., *et al.* Single-Cell Genomics Reveals a Novel Cell State During Smooth Muscle Cell Phenotypic Switching and Potential Therapeutic Targets for Atherosclerosis in Mouse and Human. *Circulation* **142**, 2060-2075 (2020).
19. Ma, W.F., *et al.* Enhanced single-cell RNA-seq workflow reveals coronary artery disease cellular cross-talk and candidate drug targets. *Atherosclerosis* **340**, 12-22 (2022).
20. Alsaigh, T., Evans, D., Frankel, D. & Torkamani, A. Decoding the transcriptome of calcified atherosclerotic plaque at single-cell resolution. *Commun Biol* **5**, 1084 (2022).
21. Grootaert, M.O.J. & Bennett, M.R. Vascular smooth muscle cells in atherosclerosis: time for a re-assessment. *Cardiovasc Res* **117**, 2326-2339 (2021).
22. Consortium, E.P. An integrated encyclopedia of DNA elements in the human genome. *Nature* **489**, 57-74 (2012).
23. Martens, J.H. & Stunnenberg, H.G. BLUEPRINT: mapping human blood cell epigenomes. *Haematologica* **98**, 1487-1489 (2013).
24. Aran, D., *et al.* Reference-based analysis of lung single-cell sequencing reveals a transitional profibrotic macrophage. *Nat Immunol* **20**, 163-172 (2019).
25. Depuydt, M.A.C., *et al.* Microanatomy of the Human Atherosclerotic Plaque by Single-Cell Transcriptomics. *Circ Res* **127**, 1437-1455 (2020).
26. Tabula Sapiens, C., *et al.* The Tabula Sapiens: A multiple-organ, single-cell transcriptomic atlas of humans. *Science* **376**, eabl4896 (2022).
27. Edsfeldt, A., *et al.* Interferon regulatory factor-5-dependent CD11c+ macrophages contribute to the formation of rupture-prone atherosclerotic plaques. *Eur Heart J* **43**, 1864-1877 (2022).
28. Lea Dib, L.A.K., Andreas Edsfeldt, Yasemin-Xiomara Zurke, Jiangming Sun, Mihaela Nitulescu, Moustafa Attar, Esther Lutgens, Steffen Schmidt, Marie W. Lindholm, Robin P. Choudhury, Ismail Cassimjee, Regent Lee, Ashok Handa, Isabel Goncalves, Stephen N. Sansom & Claudia Monaco. Lipid-associated macrophages transition to an inflammatory state in human atherosclerosis, increasing the risk of cerebrovascular complications. *Nature Cardiovascular Research* **2**, 656–672 (2023).
29. Fernandez, D.M., *et al.* Single-cell immune landscape of human atherosclerotic plaques. *Nat Med* **25**, 1576-1588 (2019).
30. Dong, R. & Yuan, G.C. SpatialDWLS: accurate deconvolution of spatial transcriptomic data. *Genome Biol* **22**, 145 (2021).

REVIEWER COMMENTS

Reviewer #1 (Remarks to the Author):

The Authors have seriously addressed the comments of this reviewer and provided suitable new material where necessary.

Reviewer #3 (Remarks to the Author):

1. Although the authors emphasized that SmartSeq-2 (SS2) technology is able to detect more genes per cell, the limited number of cells to be sequenced and the batch effect of different plates are major drawbacks of this technique. By utilizing a publicly available scRNA-seq databank of human carotid plaques (refer to Figure 4 for reviewers), the authors conducted preliminary analysis to demonstrate the identification of four major SMC subsets within plaques, thereby supporting the author's claims regarding their SMC classification. The authors may take major advantage by including this data, as an independent validation data set, to substantiate the key points of current manuscript, including the TGFB isoform expression patterns by different SMC subsets.

2. Regarding the blood contamination issues, changing the name of monocytes into macrophages did not really address this review's concern. Notably, the authors introduced a red blood cell lysis step during tissue preparation, this is the step to remove the erythrocytes from samples, meanwhile, ensuring that the blood-derived CD45+ leukocytes are retained within the samples. The authors should validate their major claims regarding CD45+ leukocytes in an independent data set, i.e., publicly available scRNA-seq of human carotid plaques, which would be suitable for this purpose.

Point by point response to reviewers

for

“Dysregulation of MMP-2-dependent TGF- β 2 activation impairs fibrous cap formation in type 2 diabetes associated atherosclerosis”

Pratibha Singh, Jiangming Sun, Michele Cavallera, Dania Al Sharify, Frank Matthes, Mohammad Barghouth, Christoffer Tengryd, Pontus Dunér, Ana Persson, Lena Sundius, Mihaela Nitulescu, Eva Bengtsson, Sara Rattik, Daniel Engelbertsen, Marju Orho-Melander, Jan Nilsson, Claudia Monaco, Isabel Goncalves, Andreas Edsfeldt

REVIEWER COMMENTS

Reviewer #1 (Remarks to the Author):

The Authors have seriously addressed the comments of this reviewer and provided suitable new material where necessary.

We are grateful for the reviewer's comments and glad that this was acknowledged.

Reviewer #3 (Remarks to the Author):

1. Although the authors emphasized that SmartSeq-2 (SS2) technology is able to detect more genes per cell, the limited number of cells to be sequenced and the batch effect of different plates are major drawbacks of this technique. By utilizing a publicly available scRNA-seq databank of human carotid plaques (refer to Figure 4 for reviewers), the authors conducted preliminary analysis to demonstrate the identification of four major SMC subsets within plaques, thereby supporting the author's claims regarding their SMC classification. The authors may take major advantage by including this data, as an independent validation data set, to substantiate the key points of current manuscript, including the TGFB isoform expression patterns by different SMC subsets.

We appreciate the reviewer's detailed comments and have now added a validation analysis, using an independent *single-cell RNA sequencing (scRNA-seq) dataset* of human carotid plaques (proximal and core regions, n=3 each group)¹ in the revised manuscript. The overview of the clustering analysis of this data set, identifying both immune and non-immune cell clusters, is provided in the new Supplementary Fig. 7 (below).

a)

b)

New Supplementary Fig. 7. UMAP visualization of all cells obtained from the PlaqView² analysis of human carotid plaques scRNA-seq data ($n=6$ plaque regions)¹. (a) Identified clusters in the PlaqView. (b) Expression of main cell marker genes.

The dataset was accessed and analyzed using the PlaqView pipeline (www.plaqview.com), where all cells were clustered. VSMCs were identified by two approaches: 1) by the expression of main differentially expressed cell marker genes and 2) by mapping the VSMCs clusters identified from our study on the above-mentioned data set. In both approaches, we could confirm the existence of the four VSMCs phenotypes in the atherosclerotic core and the proximal regions. (Supplementary. Fig. 8a and 8b).

Supplementary Fig. 8. Validation of vascular smooth muscle cell subtypes identified in the current study using an independent scRNA-seq dataset of human atherosclerotic plaques¹. (a) Expression (left) of main differentially expressed genes of VSMC subtypes recognised in current study, and their projection (right) on the VSMCs of the atherosclerotic core region (n=3) (b) and in the proximal adjacent region (n=3).

Next, the same data set was used to validate the expression of the three *TGFB* isoforms. However, when looking at the expression levels of *TGFB* genes in the two datasets, we found that the 10X platform detects the *TGFB2* gene poorly, as a low expressing gene. The maximum count of *TGFB2* detected in this dataset is 4 per cell, while the majority of cells displayed 0 counts. This makes the comparison of *TGFB2* expression between the validation dataset and our present dataset difficult (see Fig. 1 for reviewer).

Figure 1 for the reviewer. Comparing TGF isoform expressions in vascular smooth muscle cells using 10x (Alsaigh et. al. 2022)¹ and Smart-Seq2 (SS2, "Ours", our analysis) technologies).

Still, prompted by the reviewer and in an attempt to make a fair comparison, we examined the *TGFB2* gene expression across the samples to identify a region with high *TGFB2* expression. Interestingly, we found that *TGFB2* expression was lower in the atherosclerotic core region of plaques compared to proximal regions (Supplementary Fig. 9a).

New Supplementary Fig. 9a. Validation of *TGFB2* expression in a publicly available scRNA-seq dataset of human atherosclerotic plaques¹. Higher VSMC *TGFB2* expression was detected in the proximal adjacent region (PA) compared to the atherosclerotic core regions (AC) in this dataset ($n=3$ in each group). Gene expression is the \log_2 -transformed count with offset of 0.5. Gene expression of -1 denotes that 0 counts of such gene were detected.

Thus, attempting to validate our *TGFB2* finding, the dataset of VSMCs from proximal regions was used, and the contractile VSMCs did express higher *TGFB2* when compared to other VSMCs (new Supplementary Fig. 9b).

New Supplementary Fig. 9b. *TGFB2* expression was higher in the contractile VSMCs compared to other VSMCs in the proximal adjacent region (n=3). Gene expression is the log₂-transformed count with offset of 0.5. Gene expression of -1 denotes that 0 counts of such gene was detected.

In summary, using this dataset, despite the technical differences, we could to validate the four VSMC phenotypes identified in our single cell dataset of CD45⁻ plaque cells. Furthermore, even though the expression of *TGFB* isoforms is not detectable in most cells by the 10x technology, we were able to validate a greater expression of *TGFB2* in contractile VSMC.

The manuscript was revised accordingly, please see the main manuscript page 8, lines 5-12 and the supplementary material page 12-13 line 10-24:

“Additionally, we validated our main findings of the four VSMC clusters, as well as the immune cell clusters by prediction and main marker genes expression using a publicly available single cell RNA sequencing data set of human carotid plaque cells (detailed description in Supplementary methods; Supplementary Fig 7 and 8). Using this publicly available dataset, we identified higher TGFB2 expression in the proximal region (Supplementary fig. 9a), in particular in contractile VSMC compared to other VSMC (Supplementary Fig. 9b) and confirmed that TGFB2 and TGFB3 are most prominently expressed by non-immune cells (Supplementary Fig. 10)”

“Validation of the identified cell types in an independent dataset.

Publicly accessible single-cell RNA sequencing (scRNA-seq) data from human carotid plaques, encompassing both the atherosclerotic core (AC) and the proximal adjacent (PA) region, were utilized to validate our findings. This independent dataset was analysed through a reproducible pipeline designed for scRNA-seq. Based on gene expression of markers for vascular smooth muscle cells (VSMCs, Supplementary Fig. 7), the cells from cluster 5, 8, 11 and 12 were suggested as VSMCs in the independent data set. Focusing on these VSMCs, we examined the expression of main differentially expressed genes (DEGs) of the identified VSMCs clusters (namely contractile, adipocyte-like, synthetic/fibroblast-like and macrophage-like VSMCs) from our data set (Supplementary Fig. 8). Additionally, we predicted VSMC cell types by using our CD45⁻ cell types as reference. Prediction score for each cell was obtained by label transfer function of the Seurat (Supplementary Fig. 8).

Gene expression of markers for immune cells such as T cells (CD3E, CD4, CD8A), NK cells (NCAM1, NKG7, XCL1), myeloid cells (CD163, FCGR1A, ITGAX, C1QA, IL1B, CD1C), B-cells (CD79A), plasmacytoid dendritic cells (CLEC4C) and mast cells (KIT) were examined in this independent data. Based on gene expression of these markers, cells in the clusters 0, 1 and 17 (obtained from the PlaqView pipeline) were annotated as T cells, cluster 6 as NK cells, clusters 3, 4 and 7 as myeloid cells, clusters 10 and 14 as B cells, cluster 18 as plasmacytoid dendritic cells and cluster as 16 mast cells. Gene expression of markers for non-immune cells (VSMC: ACTA2, MYH11, TAGLN; fibroblast: PLA2G2A, FBLN1; endothelial cells:

PECAM1 VWF) were also examined. As previously described, cells from clusters 5, 8, 11 and 12 were annotated as VSMCs, cluster 13 as fibroblast, clusters 2, 9 and 15 as endothelial cells. Next, gene expression of TGF isoforms were examined in this independent data, split by immune cells (T, NK, Myeloid, pDC, mast) and non-immune cells (VSMC, fibroblast, endothelial cells; Supplementary Fig. 10).

Validation of the TGF isoform expression in an independent dataset

Using the same dataset as in the cell type validation, the expression of the three TGFB isoforms were assessed. When comparing the TGFB2 expression between the Smart-Seq2 (SS2) single cell analysis to the 10x single cell RNA sequencing analysis, we identified that TGFB2 expression was detected in higher levels by SS2 than by the 10x ($p < 2.2 \times 10^{-16}$). We compared the TGFB2 expression between the core and proximal regions and found that TGFB2 was detected to a greater extent in proximal regions (Supplementary Fig. 9a). These were then used to validate the expression of TGFB2 among the VSMCs. Considering the very low expression of TGFB isoforms and heterogeneity of plaques, negative binomial regressions were applied to compare the expression of TGFB2 between the predicted contractile VSMCs and rest of the VSMCs within the proximal region for each patient. A fixed-effect meta-analysis employing the inverse-variance method was performed to aggregate the differences in expression between contractile VSMCs and the other VSMCs within proximal regions. Results were shown in the Supplementary Fig. 9b”

2. Regarding the blood contamination issues, changing the name of monocytes into macrophages did not really address this review's concern. Notably, the authors introduced a red blood cell lysis step during tissue preparation, this is the step to remove the erythrocytes from samples, meanwhile, ensuring that the blood-derived CD45+ leukocytes are retained within the samples. The authors should validate their major claims regarding CD45+ leukocytes in an independent data set, i.e., publicly available scRNA-seq of human carotid plaques, which would be suitable for this purpose.

We thank the reviewer for acknowledging our effort to use the Tabula as a reference data set for the nomenclature in the revised manuscript. We agree that we cannot exclude a minor contamination of cells originating from the circulation. Notwithstanding the origin of the macrophage population, our claim is that *TGFB2* and *TGFB3* are mainly expressed by non-immune cells. To validate this claim we used the publicly available dataset by Alsaigh et. al¹. Also, in this independent dataset, we confirmed that *TGFB2* and *TGFB3* are most prominently expressed by non-immune cells, as shown in Supplementary Fig. 10.

New Supplementary. Fig. 10. Gene expression of *TGFB* isoforms in non-immune cells and immune cells. Each column represents a cell where dark blue denotes a high expression level.

This has now been added to the manuscript, please see page 8 line 9-12 and the supplementary material page 12-13 line 22-9:

*“Using this publicly available dataset, we identified higher *TGFB2* expression in the proximal region (Supplementary fig. 9a), in particular in contractile VSMC to other VSMC (Supplementary Fig. 9b) and confirmed that *TGFB2* and *TGFB3* are most prominently expressed by non-immune cells (Supplementary Fig. 10)”*

*“Gene expression of markers for immune cells such as T cells (*CD3E*, *CD4*, *CD8A*), NK cells (*NCAM1*, *NKG7*, *XCL1*), myeloid cells (*CD163*, *FCGR1A*, *ITGAX*, *C1QA*, *IL1B*, *CD1C*), B-cells (*CD79A*), plasmacytoid dendritic cells (*CLEC4C*) and mast cells (*KIT*) were examined in this independent data. Based on gene expression of these markers, cells in the clusters 0, 1 and 17 (obtained from the PlaQView pipeline) were annotated as T cells, cluster 6 as NK cells, clusters 3, 4 and 7 as myeloid cells, clusters 10 and 14 as B*

cells, cluster 18 as plasmacytoid dendritic cells and cluster as 16 mast cells. Gene expression of markers for non-immune cells (VSMC: ACTA2, MYH11, TAGLN; fibroblast: PLA2G2A, FBLN1; endothelial cells: PECAM1 VWF) were also examined. As previously described, cells from clusters 5, 8, 11 and 12 were annotated as VSMCs, cluster 13 as fibroblast, clusters 2, 9 and 15 as endothelial cells. Next, gene expression of TGF isoforms were examined in this independent data, split by immune cells (T, NK, Myeloid, pDC, mast) and non-immune cells (VSMC, fibroblast, endothelial cells; Supplementary Fig. 10)."

References:

1. Alsaigh, T, Evans, D, Frankel, D & Torkamani, A. Decoding the transcriptome of calcified atherosclerotic plaque at single-cell resolution. *Commun Biol* **5**, 1084 (2022).
2. Ma, WF, *et al.* Enhanced single-cell RNA-seq workflow reveals coronary artery disease cellular cross-talk and candidate drug targets. *Atherosclerosis* **340**, 12-22 (2022).

REVIEWERS' COMMENTS

Reviewer #3 (Remarks to the Author):

The authors have addressed this reviewer's concerns.